# Mean-field theory of 1+1D $\mathbb{Z}_2$ lattice gauge theory with matter

Matjaž Kebrič[1, 2, *] Ulrich Schollwöck[1, 2] and Fabian Grusdt[1, 2]

[1]*Department of Physics and Arnold Sommerfeld Center for Theoretical Physics (ASC),*
*Ludwig-Maximilians-Universität München, Theresienstr. 37, München D-80333, Germany*
[2]*Munich Center for Quantum Science and Technology (MCQST), Schellingstr. 4, D-80799 München, Germany*
(Dated: April 4, 2024)

Lattice gauge theories (LGTs) provide valuable insights into problems in strongly correlated many-body systems. Confinement which arises when matter is coupled to gauge fields is just one of the open problems, where LGT formalism can explain the underlying mechanism. However, coupling gauge fields to dynamical charges complicates the theoretical and experimental treatment of the problem. Developing a simplified mean-field theory is thus one of the ways to gain new insights into these complicated systems. Here we develop a mean-field theory of a paradigmatic 1+1D $\mathbb{Z}_2$ lattice gauge theory with superconducting pairing term, the gauged Kitaev chain, by decoupling charge and $\mathbb{Z}_2$ fields while enforcing the Gauss law on the mean-field level. We first determine the phase diagram of the original model in the context of confinement, which allows us to identify the symmetry-protected topological transition in the Kitaev chain as a confinement transition. We then compute the phase diagram of the effective mean-field theory, which correctly captures the main features of the original LGT. This is furthermore confirmed by the Green's function results and a direct comparison of the ground state energy. This simple LGT can be implemented in state-of-the art cold atom experiments. We thus also consider string-length histograms and the electric field polarization, which are easily accessible quantities in experimental setups and show that they reliably capture the various phases.

## I. INTRODUCTION

Lattice gauge theories (LGTs) are often studied in condensed matter systems, although they originally stem from high-energy physics [1, 2]. They are highly successful in providing physical insights into strongly correlated systems, for example, topological spin liquids or the XY-model [2]. Another signature property of the LGTs is the emergence of confinement of particles, when gauge fields are coupled to matter [3].

However, additional degrees of freedom which are introduced with LGTs makes their theoretical and experimental study complicated, since the effective Hilbert space is enlarged and one has to take into account a large set of local constraints, which arise from the gauge structure [4]. Although effective numerical techniques have been developed in recent years a better effective understanding of the lattice gauge theories is needed, especially when the system is doped by adding matter and the gauge fields become dynamical. Hence, a simplified mean-field theory which captures the essence of the LGTs is desirable to gain a better theoretical understanding.

The $\mathbb{Z}_2$ lattice gauge theory [5] has the simplest gauge structure, and has gained a lot of interest in recent years due to the direct connection to high temperature superconductivity [6–8], and its experimental implementation with cold atom setups [9, 10]. A building block of a $\mathbb{Z}_2$ lattice gauge theory has already been implemented [11, 12] using the principle of Floquet engineering [13]. Subsequently, many new proposals have been put forward including for superconducting qubits [14] and Ryd-

berg tweezer arrays [15]. To circumvent the implementation of Hamiltonians with tedious multi-body interaction terms in cold-atom setups, a powerful gauge protection scheme has been put forward [16, 17]. In addition, there has been an increased amount of proposals and implementations with digital quantum computers [18–23].

Here we study a 1+1D $\mathbb{Z}_2$ LGT where dynamical charges are coupled to gauge fields, where we generalise the problem by including superconducting (SC) terms which explicitly break the U(1) conservation of the charge number. We consider the so called physical sector without background charges which is defined by the set of local generators of the gauge structure, which is the LGT counterpart of the Gauss law [4]. Individual particles (partons) bind into dimers (mesons) in the presence of the $\mathbb{Z}_2$ electric field term in the Hamiltonian, which induces dynamics in the $\mathbb{Z}_2$ gauge field [24]. The confinement mechanism can be understood by mapping the system to the so called non-local string length basis, where confinement can be related to the breaking of the translational symmetry [25]. In addition, a smooth confinement-deconfinement crossover was uncovered as a function of temperature [26].

This model also exhibits interesting phase diagram at two-third filling, where mesons form a Mott state in the presence of the nearest-neighbor (NN) repulsion between the charges [25]. At half filling a parton Mott state stabilized by the same NN repulsion can be melted by the $\mathbb{Z}_2$ electric field term, with a plasma-like crossover region [27]. With the SC terms included, this LGT corresponds to the gauged Kitaev chain, and the corresponding phase diagram is known to exhibit a transition from the topological to the trivial state [28].

In our work we first establish the phase diagram of the

---
* matjaz.kebric@physik.uni-muenchen.de

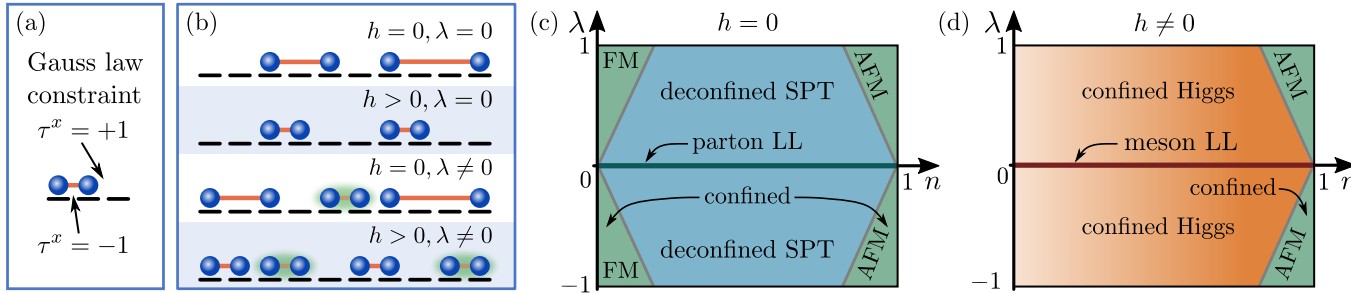

FIG. 1. One-dimensional $\mathbb{Z}_2$ lattice gauge theory with superconducting term Eq. (1). (a) The Gauss law constraint Eq. (2) ensures that the $\mathbb{Z}_2$ electric field changes its prefactor across an occupied lattice site, which allows us to define the $\mathbb{Z}_2$ strings (orange lines) labeling $\tau^x = -1$ and anti-stings (no line) which labels $\tau^x = +1$. (b) Different regimes of the 1+1D $\mathbb{Z}_2$ LGT Eq. (1). In the first row (from above) we sketch the free parton regime, in the second row we sketch the regime where partons are confined into mesons, in the third row we illustrate deconfined partons with SC term, where the U(1) symmetry is explicitly broken (meson creation and annihilation), and in the last row we illustrate the confined regime with the SC term. (c) A qualitative sketch of a phase diagram in the deconfined regime $h = 0$, which exhibits a deconfined parton Luttinger liquid (LL) on the $\lambda = 0$ line, a deconfined symmetry protected state (SPT) for intermediate fillings which correspond to $t < 2|\mu|$, ferromagnetic (FM) symmetry broken phase at low filling, and antiferromagnetic (AFM) symmetry broken phase at high filling. (d) A qualitative sketch of a phase diagram in the regime $h \neq 0$, which exhibits a confined meson LL on the $\lambda = 0$ line, confined Higgs phase for $\lambda \neq 0$ up to moderate doping, and a symmetry broken AFM phase for high doping.

1+1D $\mathbb{Z}_2$ LGT as a function of filling and the strength $\lambda$ of the SC pairing term. We considering the regime with and without the confining $\mathbb{Z}_2$ electric field term $h$ and reproduce the phase diagram of Ref. [28]. However, our analysis focuses on the microscopic picture of the doped $\mathbb{Z}_2$ LGTs; in particular we study the confinement in the presence of the SC term. In addition to entanglement entropy calculations, we also study the $\mathbb{Z}_2$ invariant Green's function and sample string-length histograms in order to probe the confinement. We show that confining features are also present in the topological trivial states at finite filling, where the $\mathbb{Z}_2$ electric field term strength is zero, $h = 0$. Notably, this establishes the symmetry-protected topological (SPT) transition in the Kitaev chain as a confinement transition, akin to the topological confinement transition known from the $2 + 1$D perturbed toric code [29–32].

We then derive a mean-field description of the $\mathbb{Z}_2$ LGT by effectively decoupling matter and $\mathbb{Z}_2$ fields, but enforcing the Gauss law on the mean-field level. We find that the mean-field analysis qualitatively captures the main features of the $\mathbb{Z}_2$ LGT aside from the U(1) symmetric critical line, since the mean-field theory explicitly breaks this U(1) symmetry. In addition, we probe the confinement in the mean-field theory and reveal qualitative agreement with the original $\mathbb{Z}_2$ LGT. Finally, we explicitly compare the ground state energies and the $\mathbb{Z}_2$ electric field polarization in the original $\mathbb{Z}_2$ LGT and its mean-field theory, where we find excellent agreement between both theories.

We use state-of-the-art DMRG calculations [33, 34] in combination with standard analytical techniques to solve the full and the mean-field models. Moreover, we explain the connection of our work to the Kitaev chain [35] obtained in the absence of gauge fields and through its mappings to spin systems [36–38]. Finally, our results are

in agreement with the previous work on gauged Kitaev chains by Borla et al. in Ref. [28], although we reach a different conclusion about the confinement of the $\mathbb{Z}_2$ charges in the limit of a static gauge field – consistent by various confinement order parameters that we analyse.

The structure of this paper is as follows: the first, introductory section I is followed by section II where we introduce the $\mathbb{Z}_2$ LGT with SC terms and explain its main features. In the third section III we present the phase diagram of the $\mathbb{Z}_2$ LGT, by calculating the entanglement entropy. We also study the confinement where we consider the $\mathbb{Z}_2$ gauge-invariant Green's function and string-length histograms. In the following section IV we present the mean-field theory of the $\mathbb{Z}_2$ LGT. We first derive the mean-field equations and then present the DMRG results of the entanglement entropy and construct the mean-field phase diagram. We also study the confinement in the mean-field theory by considering the Green's function and the string-length distributions. In section V we directly compare the mean-field theory to the original $\mathbb{Z}_2$ LGT, by considering the ground state energies and polarization of the $\mathbb{Z}_2$ electric fields. Finally we conclude and summarize our findings in the last section VI.

## II. MODEL

In this work we consider a one-dimensional (1+1D) $\mathbb{Z}_2$ lattice gauge theory, where hard-core bosons are coupled to a $\mathbb{Z}_2$ gauge field [24, 25, 27, 28]

$$\hat{\mathcal{H}} = -t \sum_{\langle i,j \rangle} \left( \hat{a}_i^\dagger \hat{\tau}_{\langle i,j \rangle}^z \hat{a}_j + \text{h.c.} \right) - h \sum_{\langle i,j \rangle} \hat{\tau}_{\langle i,j \rangle}^x$$
$$+ \lambda \sum_{\langle i,j \rangle} \left( \hat{a}_i^\dagger \hat{\tau}_{\langle i,j \rangle}^z \hat{a}_j^\dagger + \text{h.c.} \right) + \mu \sum_j \hat{n}_j. \quad (1)$$

Here $\hat{a}_j^\dagger$ ($\hat{a}_j$) is the hard-core boson creation (annihilation) operator on site $j$ and $\hat{n}_j = \hat{a}_j^\dagger \hat{a}_j$ is the corresponding number operator. Pauli matrices $\hat{\tau}_{\langle i,j\rangle}^z$ and $\hat{\tau}_{\langle i,j\rangle}^x$ defined on the links between neighboring lattice sites represent the $\mathbb{Z}_2$ gauge and electric fields, respectively.

We also consider a set of local operators [4]

$$\hat{G}_i = \hat{\tau}_{\langle i-1,i\rangle}^x \hat{\tau}_{\langle i,i+1\rangle}^x (-1)^{\hat{n}_i}, \tag{2}$$

which define the Gauss law of the $\mathbb{Z}_2$ lattice gauge theory. These operators commute on different lattice sites $[\hat{G}_i, \hat{G}_j] = 0$ and with the Hamiltonian $[\hat{G}_j, \hat{\mathcal{H}}] = 0$ [4]. The eigenvalues of Eq. (2) can take two possible values $g_j = \pm 1$ and we constrain our Hilbert space to the subsector where the eigenvalues are equal to $g_j = +1, \forall j$. The $\mathbb{Z}_2$ electric field is thus anti-aligned across an occupied lattice site and aligned across an empty site, see Fig. 1(a) and (b). Such constraint allows us to define $\mathbb{Z}_2$ electric strings and anti-strings which connect the pairs of individual particles and reflect the orientation of the electric field $\tau_{\langle i,j\rangle}^x$ [24, 25]. We define the string as $\tau^x = -1$ and the anti-string as $\tau^x = +1$, see Fig.1(a).

The first term in the Hamiltonian (1) describes the hopping of particles to their neighboring sites. In order to satisfy the Gauss law, the string has to remain attached to the particle when it hops. This is ensured by the $\hat{\tau}_{\langle i,j\rangle}^z$ operator in the hopping term. The partons are completely free in the absence of the $\mathbb{Z}_2$ electric field term $h = 0$: In this case the $\mathbb{Z}_2$ gauge fields can be eliminated by attaching strings of gauge operators to the charge operators and the $\mathbb{Z}_2$ LGT can be mapped to a free fermion model [4, 24, 28], see also Appendix A.

The third term creates or annihilates particle pairs on neighboring lattice sites

$$\hat{\mathcal{H}}_\lambda = \lambda \sum_{\langle i,j\rangle} \left( \hat{a}_i^\dagger \hat{\tau}_{\langle i,j\rangle}^z \hat{a}_j^\dagger + \text{h.c.} \right), \tag{3}$$

and can be regarded as a $\mathbb{Z}_2$ invariant superconducting (SC) term. This term explicitly breaks the U(1) symmetry of the charges and thus the particle number conservation, since it adds or removes a pair of particles while conserving the Gauss law constraint.

The $\mathbb{Z}_2$ electric field term with strength $h$ induces dynamics in the gauge fields and leads to a linear confining potential for the strings [24, 25]. It has been show that the $\mathbb{Z}_2$ lattice gauge theory in Eq. (1) without superconducting terms ($\lambda = 0$) exhibits confinement of particles into mesons for any nonzero value of the $\mathbb{Z}_2$ electric field term $h$ [24, 25]. Confined mesons remain mobile and form a meson Luttinger liquid (LL), which differs from the $h = \lambda = 0$ case where a LL of the individual partons ($\hat{a}$) forms [24, 25, 27]. The distinct nature of the two LLs at $h = 0$ and $h \neq 0$ (both at $\lambda = 0$) can be observed in the doubling of the period of Friedel oscillations in the confined regime [24]. A general solution of the confinement problem in the 1+1D $\mathbb{Z}_2$ LGT without superconducting term was found by mapping the above model to a non-local string-length basis, where it has been shown that

the translational symmetry breaking in the new basis is directly related to the confinement in the original $\mathbb{Z}_2$ LGT [25]. Furthermore, it has been shown that the model features a smooth confinement-deconfinement crossover as a function of temperature [26].

Finally, the last term in the Hamiltonian proportional to $\propto \mu \sum_j \hat{n}_j$ is added to control the effective filling of the chain, which we define as $n = \frac{1}{L} \sum_{j=1}^L \langle \hat{n}_j \rangle$, where $L$ is the chain length; $\mu$ denotes a chemical potential.

In the absence of confining $\mathbb{Z}_2$ electric field terms, i.e., for $h = 0$, and at the superconducting term value $\lambda = -t$, eliminating the gauge field yields the analytically most easily accessible limit of the Kitaev chain [35], see Appendix A. The latter can be formally mapped to a transverse field Ising model [36]. Such system has a topologically non-trivial phase for $t/|\mu| < 2$ [35].

The full Hamiltonian of the $\mathbb{Z}_2$ LGT with superconducting terms Eq. (1), resembles a gauged Kitaev model, which was studied by Borla et al. in Ref. [28]. There, the authors focused on gauging the Kitaev chain at $\lambda = -t$ and found interesting topological phases as a function of the chemical potential $\mu$, and discussed the so called gentle gauging of the Kitaev chain. The phases that they found are in agreement with our results, but we focus on the dependency on filling $n$ and consider freely tunable $\lambda \neq \pm t$ and $h$. We summarize our results in a sketch of the phase diagram in Fig. 1(c) for $h = 0$ and in Fig. 1(d) for $h \neq 0$.

Finally, we note that rich phase diagrams can be obtained as a function of filling already without superconducting terms by adding a repulsive nearest-neighbor (NN) interaction between charges to the $\mathbb{Z}_2$ LGT, Eq. (1), for $\lambda = 0$. On the one hand, the combination of the electric field term and the NN repulsion between charges stabilizes a Mott state of confined mesons at the two-thirds filling, $n = 2/3$ [25]. On the other hand, at half filling $n = 1/2$, the Mott state of individual partons is stabilized by the NN interactions and is destroyed by applying the electric field term [27]. There the crossover regime between the parton Mott state and the meson LL exhibits pre-formed parton plasma features.

## III. PHASE DIAGRAM OF THE LATTICE GAUGE THEORY

### A. Numerical calculations

In order to obtain the ground state of the $\mathbb{Z}_2$ LGT Eq. (1) we use DMRG [33, 34]. To be more precise we use the matrix-product states (MPS) toolkit SyTen [39, 40]. By using the Gauss law constraint, where we consider the physical subspace without background charges, i.e., $g_j = +1, \forall j$ [4], we exactly map the original Hamiltonian Eq. (1) to a pure spin-1/2 model [24–27], see Appendix B for details. In this mapping, we effectively integrate out the charge degrees of freedom and directly simulate the $\mathbb{Z}_2$ gauge fields. We consider open boundary conditions

(OBC) in our calculations where we start and end our chain with a link. Hence, the total system size (of the spin chain) is $L' = L+1$ where $L$ is the number of matter lattice sites. If not stated otherwise, we consider systems with the chain length of $L+1 = 97$, i.e. $L = 96$ matter lattice sites. We limit the DMRG bond dimension to $\chi = 1024$, although the typical bond dimensions needed to achieve convergence are much smaller, i.e. the bond dimension of the ground state is typically much lower.

### B. Entanglement entropy

#### 1. Value of the entanglement entropy

To determine the phase diagram of the model, we first consider the entanglement entropy $S(x)$ as a function of filling $n$, and the strength of the SC term $\lambda$ at different $\mathbb{Z}_2$ electric field strengths $h$. To calculate the entanglement entropy we cut our chain into two subsystems $A$ and $B$, where we denote the site of the cut with $x$. The length of the subsystem $A$ is thus $x$ and the length of the subsystem $B$ is $L+1-x$. Note that the total spin system contains $L+1$ spins, since we simulate the gauge fields directly. The cuts are thus between links $\tau_{x-1,x}$ and $\tau_{x,x+1}$, i.e, on the matter site $x$.

We consider the entanglement entropy at $S(x = L/2)$ and present the result in a heat map in Fig. 2. The system exhibits symmetric features around half filling $n = 0.5$ in the absence of the $\mathbb{Z}_2$ electric field term, $h = 0$, where we observe a sharp decrease of the entanglement entropy for low and high fillings when $\lambda \neq 0$, see Fig. 2(a). We attribute this to the underlying particle-hole symmetry, and the sharp drop of the entanglement entropy to the transition at $\mu = \pm t/2$ in the Kitaev chain [35].

The particle-hole symmetry is explicitly broken when $h \neq 0$, which is reflected in the numerical results where the symmetry around half-filling $n = 0.5$ disappears for any nonzero value of $h$, see Fig. 2(b). In addition, we observe that the entanglement entropy gradually increases with increasing filling. A rapid decrease of the entanglement entropy for high enough filling is retained.

Another interesting feature in Fig. 2 is the clear distinction between $\lambda > 0$ and $\lambda < 0$, which can be seen in both diagrams. We observe a large plateau of substantial entanglement entropy in the absence of the $\mathbb{Z}_2$ electric field term ($h = 0$) for $\lambda > 0$, see Fig. 2(a). This plateau narrows for stronger values of $\lambda$ indicating that there is a phase transition as a function of filling at very high and low fillings. In contrast, there is a large plateau of almost zero entanglement entropy for $\lambda < 0$. We observe some non-zero entanglement entropy features again at low and high fillings. These features again narrow in a similar way as the plateau for $\lambda > 0$ which indicates that the same phase transition occurs also for $\lambda < 0$. Indeed, by a gauge transformation $\hat{a}_j \rightarrow e^{i\frac{\pi}{2}}\hat{a}_j$, the model Eq. (1) is equivalent at $\pm\lambda$, but this gauge transformation has a non-trivial effect on the entanglement we calculate after

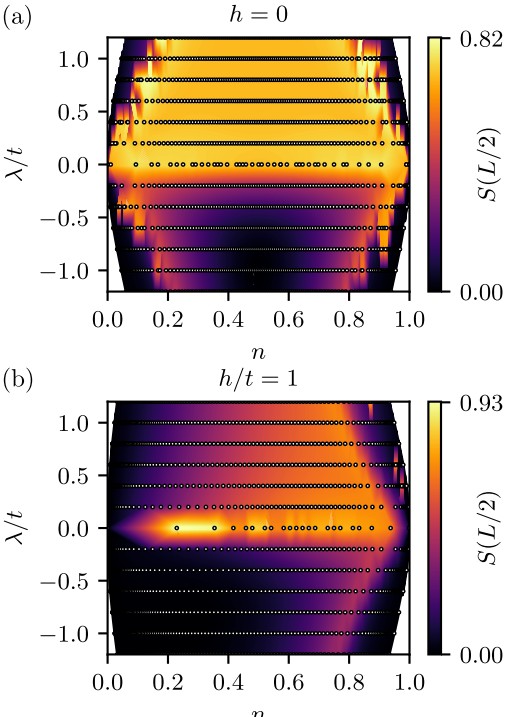

FIG. 2. Entanglement entropy of the $\mathbb{Z}_2$ LGT Eq. (1), after integrating out matter fields, as a function of filling $n$ and value of the SC term $\lambda$. (a) Symmetric behavior $(n = 0.5 + \Delta n) \leftrightarrow (n = 0.5 - \Delta n)$ is observed for $h = 0$. The entanglement entropy is larger for positive values of the SC term $\lambda > 0$ than for their negative values. We notice a trapezoid shape where the entanglement entropy drastically decreases at low filling $n \lesssim 0.15$ and high filling $n \gtrsim 0.85$, which corresponds to the transition between the topological phase and trivial phase in the Kitaev chain. (b) The entanglement entropy gradually increases with increasing filling $n$ at finite field term $h/t = 1.0$. The white dots represent the data points, obtained from DMRG, from which the software triangulates the heat map values.

integrating out matter fields, i.e., directly in the $\hat{\tau}^x_{\langle i,j\rangle}$ basis, see Appendix B.

Similar behavior in terms of $\lambda$ can be observed also for nonzero value of the $\mathbb{Z}_2$ electric field term ($h \neq 0$), see Fig. 2(b). Also here we observe that the overall value of the entanglement entropy is much lower for $\lambda < 0$ in comparison to when $\lambda > 0$.

Finally, we note that the entanglement entropy is always substantial on the $\lambda = 0$ line regardless of the confining $\mathbb{Z}_2$ electric field term $h = 0, \neq 0$.

Qualitative analysis of the entanglement entropy thus reveals a critical state at $\lambda = 0$, where we see an abrupt increase of the value of the entanglement entropy. This will be shown in the next section where we reveal that the critical lines at $\lambda = 0$ correspond to gapless Luttinger liquid. Additional transition lines are also observed for low $n \lesssim 0.15$ and high filling $n \gtrsim 0.85$ as a function of $\lambda$ for $h = 0$. For nonzero value $h \neq 0$ only the transition line at high values of filling $n \gtrsim 0.85$ survives.

### 2. Central charge

In order to study the transition lines in greater detail we extract the central charge $c$ from the entanglement entropy calculations. The entanglement entropy in infinite gapped systems generally saturates to a finite value with the increase of the subsystem length $x$, and diverges close to a quantum phase transition/in quantum critical regimes [41]. The low-lying excitations at the transition point can be described by conformal field theory (CFT) where the functional dependence of the entanglement entropy on the position of the cut $x$ was calculated analytically [41–43],

$$S(x) = S_0 + \frac{c}{6} \log \left[ \left( \frac{2L'}{\pi} \right) \sin \left( \frac{\pi x}{L'} \right) \right]. \quad (4)$$

Here $c$ is the central charge and $S_0$ is a non-universal constant. Since our calculations suffer from boundary effects due to OBC we mitigate the effect of Friedel oscillations by normalizing the entanglement entropy as [44, 45]

$$\tilde{S}(x) = \frac{S(x)}{n(x)} n, \quad (5)$$

where $n(x)$ is the local parton density and $n$ is the total lattice filling. We extracted the central charge $c$ by fitting the Eq. (4) to the normalized entanglement entropy $\tilde{S}(x)$.

The results as a function of filling $n$ and SC term value $\lambda$ in the deconfined $h = 0$ and confined regime $h/t = 1$ are presented in Fig. 3. We again observe a particle-hole symmetric behavior for zero electric field term values $h = 0$, which disappear when the electric field term is nonzero. Moreover, we observe symmetric behavior of the central charge about $\lambda = 0$. The central charge appears to be nonzero at the same lines where we expect a phase transition from the entanglement entropy value at the middle of the chain $S(L/2)$. The region where the central charge is close to zero is where the system is gapped and thus far from quantum criticality. There the system is large enough that the entanglement entropy simply saturates to a constant value and we observe a flat profile in $S(x)$ away from the boundaries, see Appendix E.

The value of the central charge on the free parton line, i.e., when the superconducting and electric field terms are zero, is close to the expected value of $c = 1$, see Fig. 3(a). This is in agreement with the gapless Luttinger liquid which the partons form in that regime [24, 25, 28]. The central charge also remains close to $c = 1$, when the electric field term is finite $h \neq 0$, but the SC term remains zero $\lambda = 0$, see Fig. 3(b). There, partons confine into mesons and the system forms a meson Luttinger liquid [24, 25].

We also observe non-zero central charge lines as a function of filling away from $\lambda = 0$. These coincide with the sharp drops of the entanglement entropy seen in Fig. 2. For $h = 0$ we observe two symmetric lines which lie on the border of the plateau of the entanglement entropy for

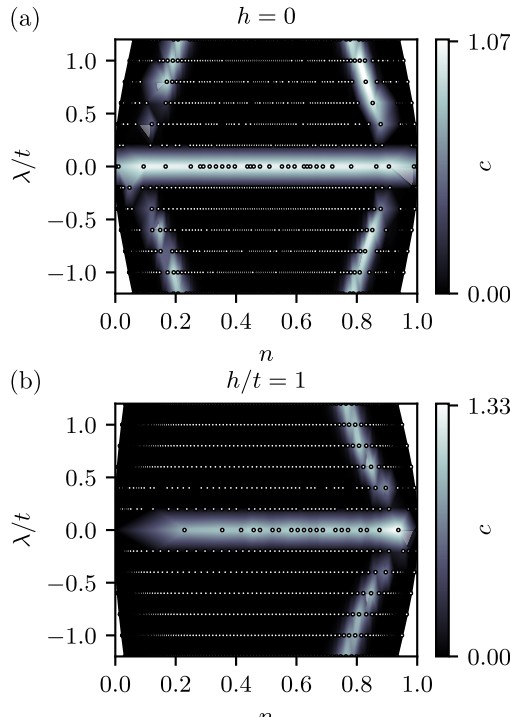

FIG. 3. Central charge as a function of filling $n$ and value of the SC term $\lambda$. (a) On the free parton line $h = 0$ and $\lambda = 0$ we extract the central charge $c = 1$, which indicates a gapless Luttinger liquid. In the case when $\lambda \neq 0$ we observe nonzero value of the central charge on the lines which correspond to the transition between the trivial and topological regime, in agreement with the results in Fig. 2. (b) In the confined regime $h/t = 1$ the $\lambda = 0$ line, where $c = 1$, remains which signals a meson Luttinger liquid. For the values $\lambda \neq 0$ we only observe one $c$ non-zero line at high filling $n \gtrsim 0.85$. The white dots represent the data points, obtained from DMRG, from which the software triangulates the heat map values.

$\lambda > 0$ and coincide with the nonzero entanglement entropy features for $\lambda < 0$. For $h = 0$ they are symmetric around $n = 0.5$.

As already stated earlier we can eliminate the gauge fields in the regime where $h = 0$ by attaching a string of gauge fields to our charge operators [4, 24]. This mapping gives us the celebrated Kitaev model for a 1D superconductor [35], see also Appendix A. Such model is known to exhibit a topological non-trivial phase for $|\mu| < 2|t|$ [35, 37], which corresponds to a finite range of fillings $0 < n_{c1} < n < n_{c2} < 1$. By plotting the same results as the function of chemical potential $\mu$, we confirm that the lines in Fig. 3(a) indeed correspond to the $|\mu| = 2t$ border between the topological trivial and non-trivial phase, see Appendix E.

The central charge at the transition between the topological and trivial state in the Kitaev chain equals to $c = \frac{1}{2}$ [46]. Our fit results slightly deviate from the exact result as the convergence becomes more involved close to a quantum phase transition. We thus often obtain fits where the central charge is slightly overestimated, how-

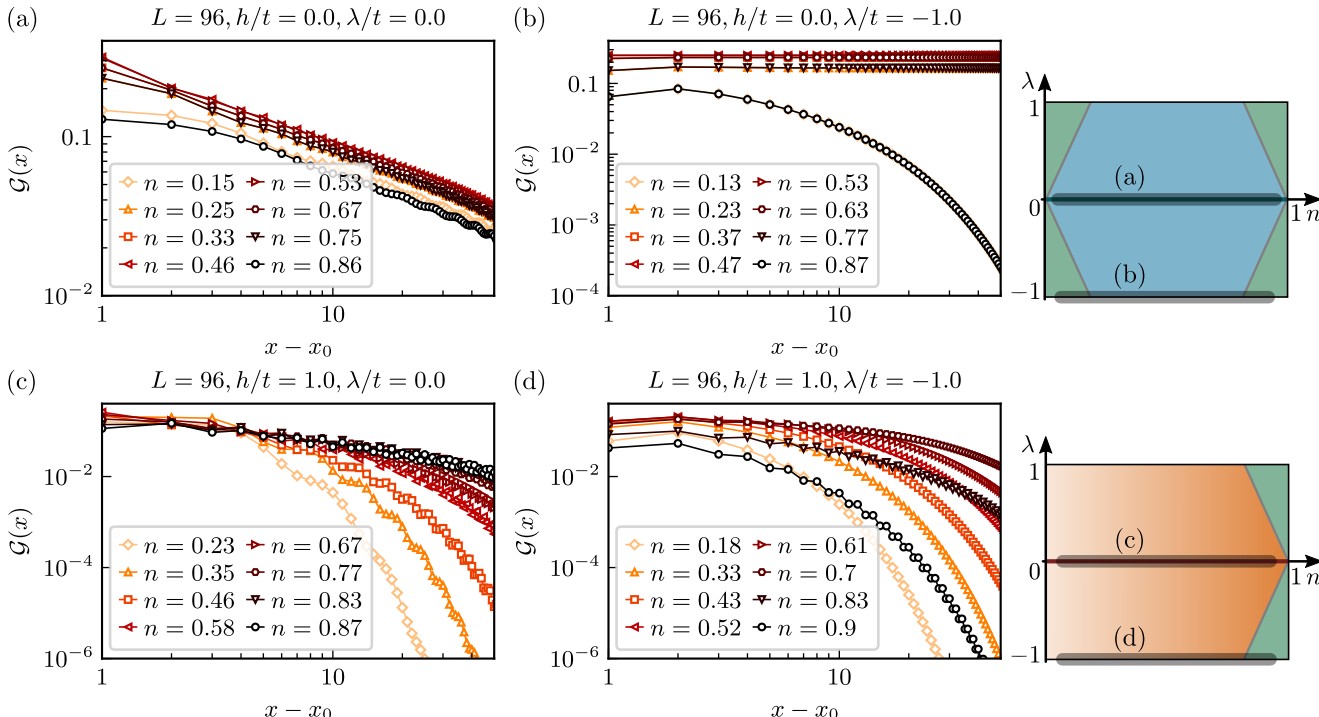

FIG. 4. Gauge-invariant Green's function Eq. (6) for the $\mathbb{Z}_2$ LGT with SC terms. (a) Free parton regime $h = 0$, $\lambda = 0$, where the Green's function decays with a power law for every filling. (b) The Green's function remains nearly constant at $h = 0$ when the SC terms are included $\lambda/t = -1$ for fillings $0.2 \lesssim n \lesssim 0.8$, which is a topologically non-trivial regime. For lower $n \lesssim 0.2$ and higher fillings $n \gtrsim 0.8$ the Green's function decays exponentially which is signaling a confined state. (c) For $h/t = 1$ and without the SC terms, $\lambda = 0$, the Green's function has an exponential decay which gets weaker with higher filling. (d) When $\lambda/t = -1$ and $h/t = 1$ we observe exponential decay which is getting weaker with increasing filling until $n \approx 0.8$, when it starts to decay stronger with increasing filling. This again signals a transition to a different state. In all plots, in order to decrease the boundary effects we start at $x_0 = 30$ in the chain length of $L = 96$. On the right side we highlighted the parameter regime in the phase diagrams where we considered the Green's functions.

ever we observe that the result are generally smaller than $c = 1$.

We conclude that the model at $h = 0$ exhibits a topological non-trivial phase inside the horizontal $c = 1$ and diagonal $c \approx 0.5$ lines, see Fig. 3(a). The area appears to decrease for larger $|\lambda|$ as a function of the filling $n$. These results agree with the arguments from Ref. [28] that the SC term opens a gap and that the two gaped regimes at $\lambda > 0$ and $\lambda < 0$ form two distinct symmetry protected topological (SPT) phases. Furthermore, the qualitative difference between the entanglement entropy in Fig. 2 supports the claim that the SPT for $\lambda > 0$ and $\lambda < 0$ are not the same [28].

Next we consider the regime with non-zero $\mathbb{Z}_2$ electric field term, $h \neq 0$, presented in Fig. 3(b). For $\lambda \neq 0$ only one $c \approx 0.5$ transition line can be observed for high filling $n$, which again matches with the entanglement entropy results in Fig.2(b) We attribute the loss of the transition line at low filling to the broken particle-hole symmetry in the $h \neq 0$ case.

We thus obtain trivial states above and below the $\lambda = 0$ line for $h/t = 1$ [28]. The transition line that remains at high doping signals the boundary between the Higgs state and the symmetry broken phase as ar-

gued in [28]. To be more precise at $\lambda = -1$ the LGT Eq. (1) reduces to an Ising model with transverse and longitudinal fields, when mapped to a spin-1/2, model in the physical Gauss sector, see Appendix B. The chemical potential $\mu$ takes the role of the Ising interaction. The symmetry broken phase at large positive $\mu > 0$ is thus the spontaneous symmetry-breaking antiferromagnetic (AFM) phase of the $\mathbb{Z}_2$ electric field, which remains stable in the anti-ferromagnetic Ising case for low values of both fields [47]. In contrast, the ferromagnetic (FM) phase at $\mu < 0$ vanishes in the presence of the longitudinal field $h \neq 0$, see also the discussion in Sec. IV B. In the LGT language a FM state corresponds to an empty chain and the AFM state to a fully filled chain.

## C. Confinement

Now we turn to a discussion of confinement in the ground state of the $\mathbb{Z}_2$ LGT, Eq.(1). We analyze different probes of confinement of dynamical charges, based on generalization of Wilson loops/Green's functions, and experimentally more accessible probes based on snapshots.

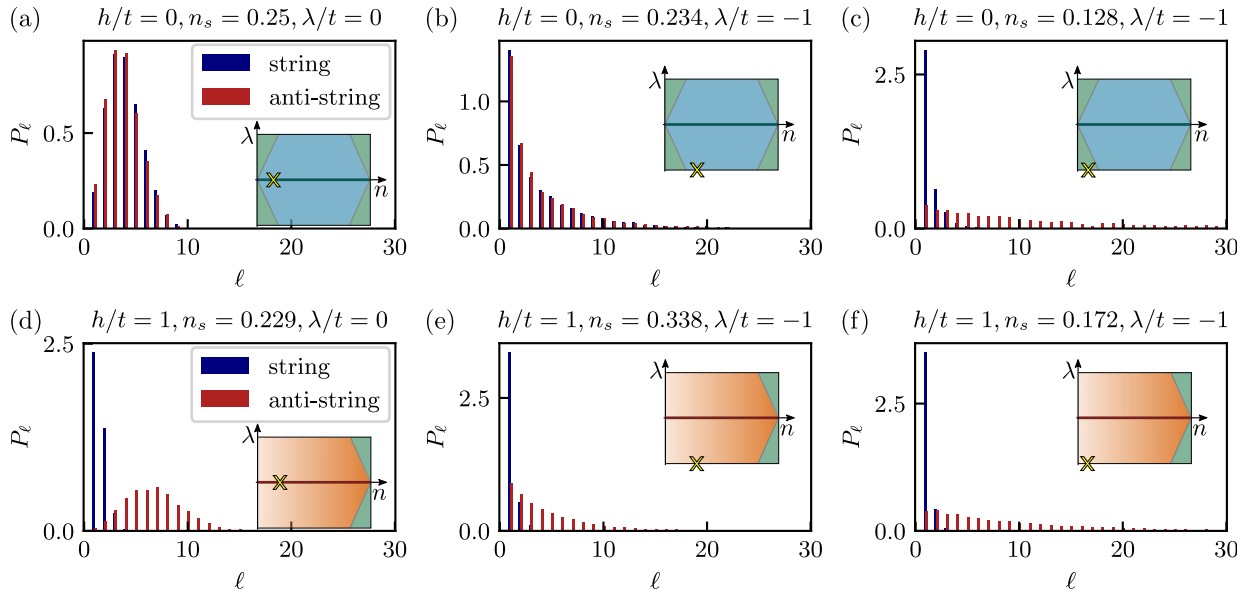

FIG. 5. String and anti-string length distributions in the 1+1D $\mathbb{Z}_2$ LGT with SC terms Eq. (1). (a) In the regime $h = \lambda = 0$ we obtain the same distributions for the string and anti-string length histograms with peaks at approximately $\ell = 3$ for quarter filling $n_s = 1/4$. (b) By including the SC terms $\lambda/t = -1$ in the deconfined SPT phase $h = 0$ at filling $n_s = 0.234$, we also obtain identical distributions of strings and anti-strings, however with a peak at $\ell = 1$. (c) For lower filling at $n_s = 0.128$ in the FM phase ($h = 0$ and $\lambda/t = -1$) we see a stronger peak at $\ell = 1$ for the strings which indicates confinement of partons into mesons. (d) In the U(1) confined phase $h/t = 1$, $\lambda = 0$, we observe two distinct distributions for strings and anti-strings, which is an indication of confinement. (e) For $h/t = 1$ with SC terms $\lambda/t = -1$ and at filling $n_s = 0.338$, both distributions peak at $\ell = 1$, however the string length distribution has a significantly higher peak than the broader anti-string length distribution, which signals confinement. (f) No qualitative change of behavior can be seen for the results in the same parameter regime as in (e) but with lower filling $n_s = 0.172$. The yellow "x" in all insets indicates the parameter regime in the corresponding phase diagram, and $n_s$ denotes the filling.

### 1. Green's function

We probe the confinement of partons into mesons by considering the $\mathbb{Z}_2$ invariant Green's function defined as [24, 25]

$$\mathcal{G}(x) = \left\langle \hat{a}_{x_0}^\dagger \Big( \prod_{x_0 \leq \ell < x} \hat{\tau}_{\ell,\ell+1}^z \Big) \hat{a}_x \right\rangle. \tag{6}$$

In the absence of the SC terms it decays as a power-law in the deconfined regime $h = 0$ and exponentially in the confined regime $h \neq 0$. We note that $\mathcal{G}(x)$ can be viewed as a one-dimensional version of the Fredenhagen-Marcu order parameter considered in higher dimensions [48]. In a deconfined phase with free partons (confined phase with partons bound into mesons) the Green's function decays with a power-law (exponentially) in $1 + 1$D dimensions.

We calculate the Green's function for different fillings and values of $h$ and $\lambda$. For the free partons $h = \lambda = 0$, we find the expected power-law decay, see Fig. 4(a) and the decay rate appears to be similar across different values of $n$. This changes for the non-zero value of the electric field term $h = 1$, $\lambda = 0$, where a clear exponential decay can be observed because the partons bind into mesons, see Fig. 4(c). This decay is slower at higher filling, where the mesons become less mobile and any hopping is restricted by other charges in the chain.

In the presence of a SC term $\lambda/t = -1$ the regime with $h = 0$ has a slightly different behavior. The Green's function remains almost constant for $0.2 \lesssim n \lesssim 0.8$, but then decays exponentially at fillings $n \lesssim 0.2$ and $n \gtrsim 0.8$, see Fig. 4(b). From the exact mapping to a transverse-field Ising model, see Appendix A, we know that the regimes of high and low densities are symmetry broken regions where the $\mathbb{Z}_2$ electric field is spontaneously ordered, $\langle \hat{\tau}_{\langle i,j \rangle}^x \rangle \neq 0$. At low (high) filling the ordering is (anti-)ferromagnetic, see also Section V with the numerical result for the polarization of the $\mathbb{Z}_2$ electric field, that corresponds to an order parameter for this symmetry breaking.

Hence the Green's function signals a regime where fluctuations are constituted by pairs of partons, thereby confining them. We note that close to the transition at low/high fillings convergence was generally harder and it is thus hard to establish at which precise filling does the transition occur [1].

When the electric field term is included, $h/t = 1$, the Green's function decays exponentially across different fill-

---

[1] However, converged data can be clearly identified as the curves are equal for fillings $n \leftrightarrow 1 - n$ due to the particle-hole symmetry when $h = 0$.

ings, and it increases its magnitude with increasing filling, see Fig. 4(d). However, this trend lasts only up until $n \approx 0.8$. For fillings $n \gtrsim 0.8$ the decay again becomes stronger. As in the case $h = 0$, we expect this behaviour of the Green's function to be related to the spontaneously broken $\mathbb{Z}_2$ symmetry which leads to long-range AFM order of $\langle \hat{\tau}^x_{\langle i,j \rangle} \rangle$ for large densities $n \gtrsim 0.8$ when $\lambda \neq 0$, irrespective of $h$.

### 2. String length histograms

We also consider string and anti-string length histograms, which we obtain by sampling snapshots from the MPS representing our ground states [26, 39, 40, 49]. We sample 400 snapshots for each data set and calculate the lengths of strings ($\tau^x = -1$) and anti-strings ($\tau^x = +1$) [26]. In a deconfined (confined) phase these histograms should coincide (differ significantly), indicating how partons connect to one another via $\mathbb{Z}_2$ electric strings. This provides a geometric picture of confinement, which has been recently generalized to higher dimensions by analyzing percolation of $\mathbb{Z}_2$ electric strings [32].

We plot the string and anti-string length distributions for different fillings $n$ and values of $h$ and $\lambda$ in Fig. 5. We observe no difference between the distributions of strings and anti-strings in the regime $h = 0$, without SC term $\lambda$, see Fig. 5(a). The distribution has a clear peak at a finite filling $\ell \approx 3$. This indicates a deconfined regime, in agreement with our results from the Green's function analysis. By including the SC term the distributions change and the string-length histograms peak at $\ell = 1$, see Fig. 5(b) and (c), which we attribute to the loss of the U(1) symmetry and the associated increase in pair number fluctuations induced by the SC term.

We note that for non-zero SC term, $\lambda/t = -1$, the string and anti-string length distributions differ for low fillings $n \lesssim 0.2$, where the string-length peak at $\ell = 1$ is much higher while the anti-string length distribution is significantly broader, see Fig. 5(c). This confirms the conclusion from our Green's function analysis in Fig. 4(c) that there exists a spontaneously confined phase for low $n \lesssim 0.2$ and high $n \gtrsim 0.8$ fillings when the SC term is non-zero $\lambda \neq 0$ and $h = 0$. The microscopic picture can also be easily described: At such low fillings all particles come from pair fluctuations generated by the SC terms. All particles thus come in pairs, i.e. in forms of mesons. The mesons are then soon annihilated, on average much sooner than the parton would be able to hop, hence they become confined.

In the spin model, which we simulate with DMRG, the picture is even more clear. For simplicity we can consider the case when $\lambda = -t$, where the model becomes a simple transverse field Ising model, see Appendix B. There, mesons are simple excitations in a form of flipped spins in the ferromagnetic state realized at $\mu < 2$ (i.e. $n \leq 0.2$) and $h = 0$, see also the discussion in Sec. IV.

In the regime $h/t = 1$ without SC terms $\lambda = 0$, depicted in Fig. 5(d), we observe two distinct distributions for the string and anti-string length histograms, respectively. The string length distribution peaks at $\ell = 1$ and the anti-string length distribution peaks at a finite value of $\ell$ which depends on the filling. The higher the filling the lower the average anti-string length in the confined regime. Such bi-modal distribution is a hallmark signature of confinement [26]. By including the SC term the anti-string length distribution narrows, see Fig. 5(d) and (f), but remains significantly wider than the string-length distribution. This is a clear indication of confinement.

We thus conclude that the string-length distributions are a good measure of confinement, fully in agreement with the Green's function analysis. In contrast to the latter, string-lengths can be directly probed in quantum simulation setups. We note that string-length distributions work well for low fillings $n < 0.5$. For higher fillings the average distance between mesons and partons in the confined and deconfined regime become similar and it is thus harder to distinguish bimodal distributions, i.e., experiments in this regime will need to acquire more data.

### D. Phase diagram

By taking into account all of the results we sketch the phase diagrams for $h = 0$ and $h \neq 0$ in Fig. 1(c) and (d). When the $\mathbb{Z}_2$ electric field term is zero, $h = 0$, the model reduces to a simple superconducting model, which exhibits a SPT phase which has a transition to trivial spontaneous symmetry broken (SSB) states [28] at low and high fillings. In the $\mathbb{Z}_2$ LGT language we obtain deconfined partons in the SPT phase, which form the parton Luttinger liquid when the U(1) symmetry is conserved on the $\lambda = 0$ line sketched in Fig. 1(c), and confined mesons in the topologically trivial SSB states.

Next we discuss the phase diagram as a function of the confining $\mathbb{Z}_2$ electric term $h$ and SC term $\lambda$ at different fillings which we present in Fig. 6. We contrast the resulting phase diagram to the phase diagram of the Fradkin-Shenker (perturbed toric code) model in Fig. 6(d) which features a topologically non-trivial deconfined phase surrounded by a confined Higgs phase [30], see also [8, 31].

At low filling, see Fig. 6(a), we find a similar structure in our 1D model, with a deconfined symmetry-protected topological phase at small $\lambda$ transitioning to a confined phase at large $\lambda$, for $h = 0$. In contrast to the 2D case, the deconfined topological phase is only robust at $h = 0$ and gives way for a confined Higgs phase when $h \neq 0$. Furthermore, the confined phase at $h = 0$ is associated with a spontaneously broken symmetry (ferromagnetic ordering of $\mathbb{Z}_2$ electric field $\hat{\tau}^x$). The latter continuously evolves into a paramagnetic state of $\mathbb{Z}_2$ electric field when $h \neq 0$ and remains confined. Finally, on the $\lambda = 0$ line for $h \neq 0$ a confined meson Luttinger liquid is realized, which transitions to a deconfined parton Luttinger liquid at the special point $\lambda = h = 0$.

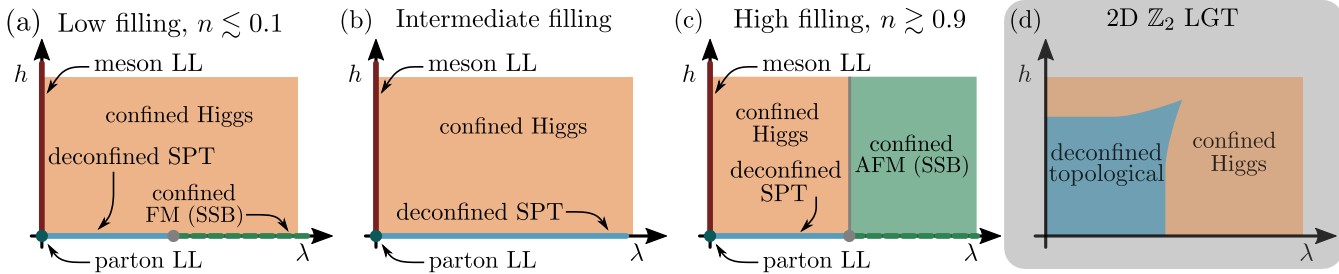

FIG. 6. Sketches of phase diagrams of the $\mathbb{Z}_2$ LGT as a function of SC term $\lambda$ and $\mathbb{Z}_2$ electric field term $h$, for increasing matter densities. (a) Free partons form a Luttinger liquid when $h = 0$ and $\lambda = 0$, which turns into meson LL for any non-zero value of $h \neq 0$ and if $\lambda = 0$. On the other hand the $\lambda \neq 0$ term explicitly breaks the U(1) symmetry and the parton LL becomes a deconfined SPT at $h = 0$, which at low filling has a transition point to a confined FM state, with spontaneous symmetry breaking (SSB). For any non-zero $h \neq 0$ and $\lambda \neq 0$, the system is in a confined Higgs state. (b) At intermediate fillings the phase diagram is similar to (a) with a difference that the system remains in a deconfined SPT phase at $h = 0$, and doesn't transition to a confined state. (c) At high filling a similar transition as in (a) occurs as a function of $\lambda$ for $h = 0$. The only difference is that in this regime the SSB state is an AFM state and that it remains stable also for $h \neq 0$. (d) For comparison we also present the phase-diagram of the $2 + 1D$ $\mathbb{Z}_2$ LGT as proposed by Fradkin and Shenker [30].

At the highest fillings the picture is similar, except that the confined, spontaneously symmetry-broken phase at $h = 0$, which features antiferromagnetic order of the $\mathbb{Z}_2$ electric fields $\langle \hat{\tau}^x_{\langle j,j+1,\rangle} \rangle = (-1)^j$, is stable upon introducing $\mathbb{Z}_2$ electric field term $h \neq 0$, see Fig. 6(c).

Finally, for intermediate densities the spontaneously symmetry-broken phase disappears and the phase diagram only contains an extended confined Higgs phase at $h, \lambda \neq 0$, in addition to the deconfined meson (parton) Luttinger liquid at any $h \neq 0$ (at the special point $h = 0$) when $\lambda = 0$.

## IV. MEAN-FIELD THEORY

Mean-field theories based on the slave-particle approach give important insights into problems in strongly correlated systems, for example, frustrated quantum magnets or the Kondo problem [50]. The main idea behind the slave-particle mean-field theories is to extend the Hilbert space and impose constraints on the mean-field level only. Such approach is thus suited to tackle the $\mathbb{Z}_2$ lattice gauge theories, by decoupling matter from gauge field, while enforcing the Gauss law on the mean-field level.

### A. Derivation of the mean-field theory

Now we develop an effective mean-field description of the $\mathbb{Z}_2$ LGT, by making the following product ansatz

$$|\psi\rangle = |\psi_\tau\rangle \otimes |\psi_a\rangle, \qquad (7)$$

where we effectively decouple the gauge ($\hat{\tau}$) and charge ($\hat{a}$) degrees of freedom. In addition, we enforce the Gauss law on the mean-field level. We can thus write down two effective Hamiltonians, one for the matter fields and the second one for the $\mathbb{Z}_2$ gauge fields.

The mean-field matter Hamiltonian can be derived by considering the link (gauge and electric) fields on the mean-field level (see Appendix C for more details), which yields

$$\hat{\mathcal{H}}^a_{\mathrm{MF}} = -t \left\langle \hat{\tau}^z_{\langle i,j \rangle} \right\rangle \sum_j \left( \hat{a}^\dagger_{j+1} \hat{a}_j + \hat{a}^\dagger_j \hat{a}_{j+1} \right)$$
$$+ \lambda \left\langle \hat{\tau}^z_{\langle i,j \rangle} \right\rangle \sum_j \left( \hat{a}^\dagger_{j+1} \hat{a}^\dagger_j + \hat{a}_j \hat{a}_{j+1} \right)$$
$$- h \left\langle \hat{\tau}^x_{\langle i,j \rangle} \right\rangle (L+1) + \mu_a \sum_j \left( \hat{a}^\dagger_j \hat{a}_j - n \right), \quad (8)$$

where the $\langle \hat{\tau}^z_{\langle i,j \rangle} \rangle = \langle \psi_\tau | \hat{\tau}^z_{\langle i,j \rangle} | \psi_\tau \rangle$ and $\langle \hat{\tau}^x_{\langle i,j \rangle} \rangle = \langle \psi_\tau | \hat{\tau}^x_{\langle i,j \rangle} | \psi_\tau \rangle$ are the averaged value of the $\mathbb{Z}_2$ gauge and electric fields, respectively. In addition, we added the Lagrange multiplier $\mu_a$ in order to enforce the correct particle filling, i.e., a chemical potential.

Note that this is a superconducting quantum wire model [35] with prefactors modified by the mean-field values of the $\mathbb{Z}_2$ gauge field and we can discard the $\mathbb{Z}_2$ electric field term which becomes a constant energy offset. This model can be solved using the Jordan-Wigner and Bogoliubov transformations [38]. By setting $\lambda = 0$ we get a simple free fermion model and by setting the $\lambda = -t$ we obtain the Kitaev model [35, 36].

Similarly, we can take the above ansatz Eq. (7) and consider the charge operators on a mean-field level which gives us a purely spin model, (see Appendix C for more details)

$$\hat{\mathcal{H}}^\tau_{\mathrm{MF}} = -t \sum_j \left( \left\langle \hat{a}^\dagger_{j+1} \hat{a}_j \right\rangle + \left\langle \hat{a}^\dagger_j \hat{a}_{j+1} \right\rangle \right) \hat{\tau}^z_{j,j+1}$$
$$+ \lambda \sum_j \left( \left\langle \hat{a}^\dagger_{j+1} \hat{a}^\dagger_j \right\rangle + \left\langle \hat{a}_j \hat{a}_{j+1} \right\rangle \right) \hat{\tau}^z_{j,j+1}$$
$$- h \sum_{\langle i,j \rangle} \hat{\tau}^x_{\langle i,j \rangle} + \mu_\tau \sum_{\langle i,j,k \rangle} \left( \hat{\tau}^x_{\langle i,j \rangle} \hat{\tau}^x_{\langle j,k \rangle} - 1 - 2n \right). \quad (9)$$

Here the Lagrange multiplier $\mu_\tau$ again ensures the correct filling and comes from the conservation law derived directly from the mean-field Gauss law,

$$\left\langle \sum_{\langle i,j,k \rangle} \hat{\tau}^x_{\langle i,j \rangle} \hat{\tau}^x_{\langle j,k \rangle} \right\rangle = L\left(1 - 2n\right), \qquad (10)$$

where $\langle i,j,k \rangle$ denotes a sequence of three consecutive lattice sites. We note that the model Eq. (9) is an Ising model with transverse and longitudinal fields. The $\mathbb{Z}_2$ electric field corresponds to the longitudinal field and the transverse field is proportional to hopping $t$ and SC term $\lambda$, normalized by the mean-field values of the charge hopping and pairing terms.

The two equations give us the full mean-field theory for charges and $\mathbb{Z}_2$ fields. We will focus on the $\mathbb{Z}_2$ field model Eq. (9) which can be written as

$$\hat{\mathcal{H}}_{\mathrm{MF}} = -g \sum_{\langle i,j \rangle} \hat{\tau}^z_{\langle i,j \rangle} - h \sum_{\langle i,j \rangle} \hat{\tau}^x_{\langle i,j \rangle} + \mu_\tau \sum_j \hat{\tau}^x_{\langle i,j \rangle} \hat{\tau}^x_{\langle j,k \rangle}, \qquad (11)$$

where we define

$$g = t\left(\left\langle \hat{a}^\dagger_{j+1} \hat{a}_j \right\rangle + \left\langle \hat{a}^\dagger_j \hat{a}_{j+1} \right\rangle\right) \\ - \lambda\left(\left\langle \hat{a}^\dagger_{j+1} \hat{a}^\dagger_j \right\rangle + \left\langle \hat{a}_j \hat{a}_{j+1} \right\rangle\right). \quad (12)$$

In order to simulate the above model using DMRG we have to compute the prefactor $g$ in front of the transverse field term, which consists of the average value of the terms in the superconducting model Eq. (8). In other words: we need to determine the average ground state energy of the superconducting model per lattice site, for given parameter values $t$, $\lambda$, and filling $n$. By diagonalizing the matter mean-field model Eq. (8), where we make use of the Jordan-Wigner and Bogoliubov transformations (see also Appendix C for more details), we obtain

$$g = \frac{1}{2\pi} \int_0^\pi \mathrm{d}k \sqrt{\left(\tilde{\mu}_a - 2t\cos(k)\right)^2 + \left(2\lambda \sin(k)\right)^2} \\ + \tilde{\mu}_a \left(n - \frac{1}{2}\right). \quad (13)$$

Note that we normalized the model in Eq. (8) by $\langle \hat{\tau}^z_{\langle i,j \rangle} \rangle$. As a result, the renormalized chemical potential $\tilde{\mu}_a = \mu_a / \langle \hat{\tau}^z_{\langle i,j \rangle} \rangle$ appears in Eq. (13). This makes the solution at a given filling $n$ independent of the gauge degrees of freedom, as we need to find the correct value of $\tilde{\mu}_a$ for the given hopping $t$ and SC term $\lambda$. To be more precise we need to find the correct value of $\tilde{\mu}_a$ which solves the equation

$$n = \frac{1}{2}\left(1 - \frac{1}{\pi} \int_0^\pi \mathrm{d}k \frac{\tilde{\mu}_a - 2t\cos(k)}{\sqrt{\left(\tilde{\mu}_a - 2t\cos(k)\right)^2 + \lambda^2(k)}}\right), \quad (14)$$

that comes from minimizing the ground state energy of the matter mean-field model (see Appendix C for more details).

The integration in Eq.(14) can be performed numerically for generic values of $t$ and $\lambda$. The $\lambda = 0$ limit can be calculated analytically and yields $g = \frac{\sin(\pi n)}{\pi}$, see Appendix C for more details.

Once the value of $g$ is established we run the DMRG to search for the correct value of $\mu_\tau$ which guarantees that Eq. (10) is satisfied, see Appendix D for details.

### B. Phase diagram of the mean-field theory

#### 1. The Ising model with transverse and longitudinal fields

The phase diagram of a generic Ising model with transverse and longitudinal fields is well established theoretically [47, 51–53], as well as experimentally [54]. One of the most important parameters is the prefactor of the Ising interaction in the presence of a non-zero longitudinal field. When the longitudinal field is zero, that is when $h = 0$ in the case of the mean-field theory in Eq. (11), the spin system forms an ordered state for $g/|\mu_\tau| < 1$. For negative Ising interaction $\mu < 0$ the ordered state is ferromagnetic and for positive Ising interaction $\mu_\tau > 0$ the ordered state is anti-ferromagnetic. Both ordered states are equivalent as the spins can be multiplied with a staggered sign yielding an opposite sign for the Ising interaction when the longitudinal field is zero, which is a particle-hole mapping in the parton language [51, 53].

This changes when the longitudinal field is nonzero. For the FM Ising interaction $\mu_\tau < 0$ we always obtain a disordered phase. However for the AFM Ising interaction $\mu_\tau > 0$ there exists an AFM lobe which remains stable up to $h/\mu_\tau < 2$ and $g/\mu_\tau < 1$ [47, 51]. In addition there is an Ising multi-critical point at $h/\mu_\tau = 2$ and $g/\mu_\tau = 0$. Both AFM and FM models have a transition point at the transverse field $g/\mu_\tau = 1$ in the absence of the longitudinal field $h = 0$ [47, 53]. Hence this means the transition between the AFM (FM) and disordered states in the effective model depends on the filling of the $\mathbb{Z}_2$ LGT system, which we want to describe.

#### 2. Numerical simulation of the mean-field theory

As for the 1+1D $\mathbb{Z}_2$ LGT with SC terms Eq. (1) we also simulate the mean-field theory Eq. (11) with DMRG. We first self-consistently solve the equations (13) and Eq. (14) in order to determine the value of $g$ at given $h$ and $\lambda$ for the filling $n$ which we want to simulate. The next step is to find the correct chemical potential $\mu_\tau$ which yields the correct target filling $n$ in Eq. (10). To this end we use a simple algorithm where we run the DMRG for different chemical potentials, see Appendix D.

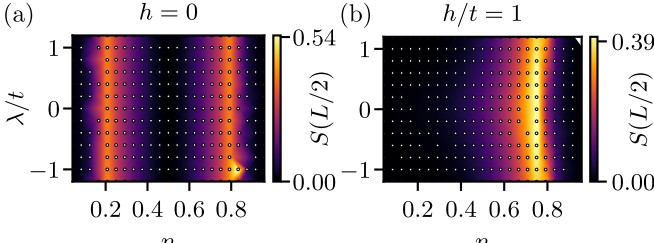

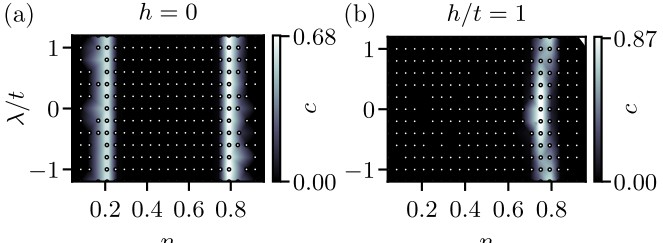

FIG. 7. Entanglement entropy in the gauge sector of the mean-field theory Eq. (11) as a function of filling $n$ and value of the SC term $\lambda$. (a) Two vertical broad bands of high entanglement entropy can be observed at fillings $n \approx 0.2$ and $n \approx 0.8$ in the regime $h = 0$. (b) Only a single vertical broad band of high entanglement entropy can be observed at filling $n \approx 0.8$ when $h/t = 1$.

FIG. 8. Central charge extracted in the mean-field theory Eq. (11) as a function of filling $n$ and the SC term $\lambda$. (a) Two vertical lines of non-zero central charge $c$ can be observed close to fillings $n \approx 0.2$ and $n \approx 0.8$ in the deconfined regime $h = 0$. (b) A single vertical line of non-zero central charge $c$ can be observed at filling $n \approx 0.8$ in the confined regime $h/t = 1$.

### 3. Mean-field theory entanglement entropy

In order to compare the mean-field theory to the full LGT we consider the entanglement entropy in the gauge sector and plot the value of the entanglement entropy when the system is cut in two equal parts $S(L/2)$ in Fig. 7. In the regime $h = 0$, we obtain two vertical lines with high entanglement entropy at fillings of $n \approx 0.2$ and $n \approx 0.8$ as a function of the SC term $\lambda$. Furthermore, when $h/t = 1$ we observe only one vertical line at high filling $n \approx 0.8$

This is qualitatively similar to the observed transition at $\lambda/t = \pm 1$ for the exact case (see Fig. 2 and Fig. 3) and comes as no surprise: The exact model reduces to an Ising model with transverse and longitudinal fields for $\lambda/t = -1$, when written in the gauge basis (integrating out matter), see Appendix B. Hence, the mean-field theory captures the transition between the confined Higgs and symmetry-broken phases.

However, there are no features visible on the $\lambda = 0$ lines in the gauge sector. We attribute this to the fact that the gauge-sector mean-field theory in Eq. (11) does not posses the U(1) symmetry in the charges, since the filling is enforced on average only, via the chemical potential $\mu_\tau$. i.e., the Lagrangian multiplier. Hence we cannot obtain the quantum criticality on the $\lambda = 0$ lines. However, the matter-sector of the mean-field theory, Eq. (8), is critical on the $\lambda = 0$ line where it conserves the global U(1) symmetry. While the mean-field ansatz overall captures free partons in the regime $h = 0$, it fails to capture in detail the confined meson Luttinger liquid at $h \neq 0, \lambda = 0$.

### 4. Central charge in the mean-field theory

We also extract the central charge from our entanglement entropy calculations. This is done in the same way as for the exact LGT where we normalized the $S(x)$ with the local filling and fit the CFT function Eq. (4) to $\tilde{S}(x)$.

The results are presented in Fig. 8 and agree with the entanglement entropy results in Fig. 7. The extracted charge is close to $c = \frac{1}{2}$ which is well-known to capture the transition in the transverse-field Ising model or, equivalently, the Kitaev chain [46].

### C. Confinement in the mean-field theory

#### 1. Mean-field theory Greens function

The next step in analyzing the mean-field theory is to consider the confinement, which is one of the most intriguing features of the $\mathbb{Z}_2$ LGT. We again start by considering the $\mathbb{Z}_2$ invariant Green's function defined in Eq. (6),

$$\mathcal{G}(x) = \left\langle \hat{a}_{x_0}^\dagger \Big( \prod_{x_0 \leq \ell < x} \hat{\tau}_{\ell,\ell+1}^z \Big) \hat{a}_x \right\rangle. \quad (15)$$

which we rewrite in the spin language by taking advantage of the Gauss law constraint to the physical sector, see Appendices B and D:

$$\mathcal{G}(x) = \left\langle \frac{1}{4} \Big( \prod_{x_0 \leq \ell < x} \hat{\tau}_{\ell,\ell+1}^z \Big) \right.$$
$$\left. \big( 1 + \hat{\tau}_{x-1,x}^x \hat{\tau}_{x,x+1}^x \big) \big( 1 + \hat{\tau}_{x_0-1,x_0}^x \hat{\tau}_{x_0,x_0+1}^x \big) \right\rangle. \quad (16)$$

In Fig. 9 (a) and (b) we observe an almost constant value of the Green's function when $h = 0$ for filling $0.2 \lesssim n \lesssim 0.8$, which corresponds to a deconfined phase. The nearly constant value can be understood from the fact that the $g\hat{\tau}^z$ term dominates the mean-field model, meaning that the spins align along the $z$-direction, which corresponds to a paramagnetic phase in the Ising model [47].

For lower $n \lesssim 0.2$ and higher $n \gtrsim 0.8$ fillings we observe an exponential decay, which coincides with the confined, symmetry broken FM and AFM phases. The qualitative behavior does not change when we include the SC term,

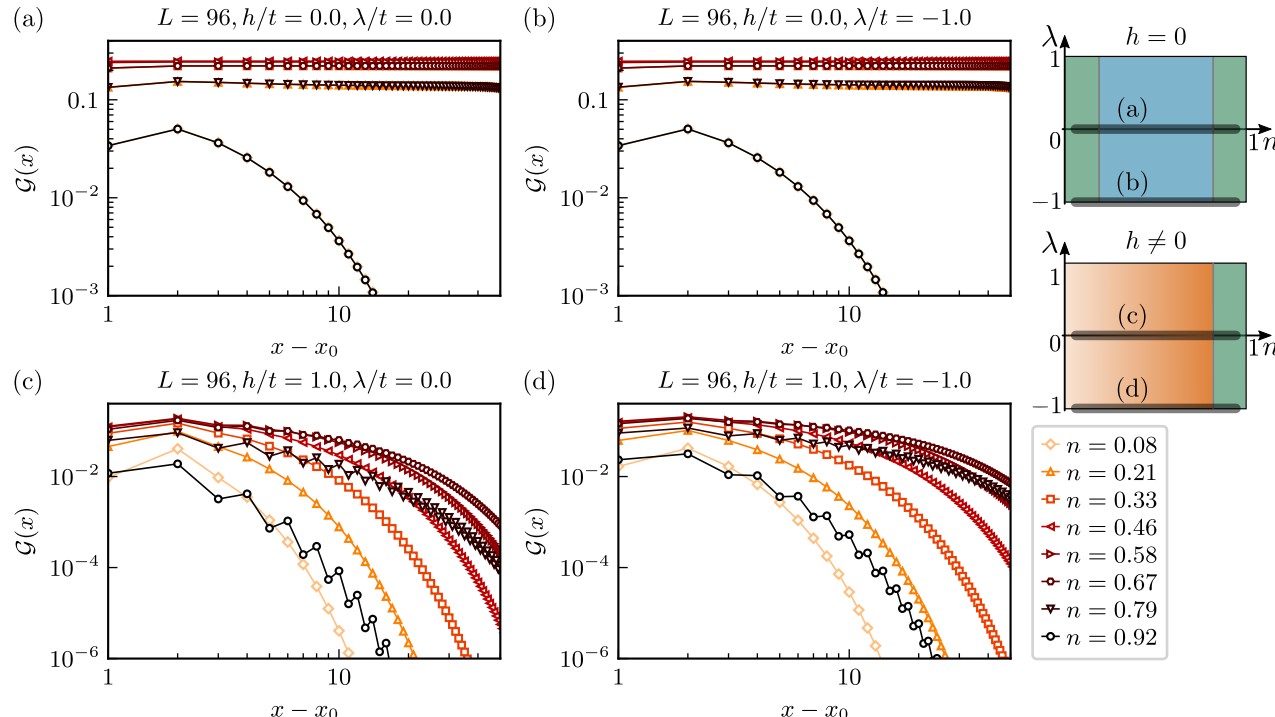

FIG. 9. Green's function Eq. (15) in the gauge sector of mean-field theory Eq. (11). (a) In the absence of the $\mathbb{Z}_2$ electric field term, $h = 0$, and the SC term, $\lambda = 0$, we obtain a nearly constant value for the Green's function for fillings $0.2 \lesssim n \lesssim 0.8$, indicating a deconfined phase. For lower and higher fillings we obtain an exponential decay in the respective confined symmetry broken FM and AFM phases. (b) We observe identical behavior as in (a) also when we include the SC terms $\lambda/t = -1$. (c) By including the $\mathbb{Z}_2$ electric field term $h/t = 1$ while setting the SC term to zero, $\lambda = 0$, we obtain an exponential decay across all fillings. (d) No qualitative difference can be observed when the SC term is also non-zero $\lambda/t = -1$ in comparison to the case in (c). On the right side we highlighted the parameter regime in the mean-field theory phase diagrams, where we considered the Green's functions. In (b) - (d) the mean-field predictions agree qualitatively with the exact LGT results.

as can be seen in Fig. 9(b). This is in line with the entanglement entropy calculations, where we found that the transition to the symmetry broken state as a function of filling does not change with the value of $\lambda$.

By including the $\mathbb{Z}_2$ electric field term, $h \neq 0$, we obtain an exponential decay across all fillings $n$, regardless of the value of the SC term $\lambda$, see Fig. 9(c) and (d). The mean-field theory in the gauge sector thus correctly captures the confined phase of the original $\mathbb{Z}_2$ LGT. Furthermore, the strength of the exponential decay decreases with increasing filling $n$ up to the filling of approximately $n \approx 0.8$ when the strength of the exponential decay starts to increase again. This behavior is exactly the same as the exponential decay in the exact LGT for $\lambda/t = -1$ in Fig. 4(d). However, this density dependence is not observed for $\lambda = 0$ in the exact LGT, where the strength of the exponential decay monotonically decreases with filling. Indeed, the mean-field theory predicts transitions to symmetry-broken states at critical fillings $n_c \approx 0.2, 0.8$ for $\lambda = 0$ which does not exists at $\lambda = 0$ in the full $\mathbb{Z}_2$ LGT model.

We thus conclude that the mean-field theory captures the main features of the Green's function behavior for different values of the $\mathbb{Z}_2$ electric field for $\lambda \neq 0$. For $h = 0$

it also captures the confinement-deconfinement transition to the spontaneously symmetry breaking phases at low and high fillings. However, such transition is also observed for $\lambda = 0$ since there is no U(1) symmetry in the mean-field theory in the gauge sector.

### 2. Mean-field theory string-length distributions

Next we consider string and anti-string length distributions in the mean-field theory. As can be seen in Fig. 10, the distributions peak at $\ell = 1$ for every parameter regime. However, the string-length peaks are significantly higher, and the anti-string length distribution is significantly broader, when $h/t = 1$ in Fig. 10(d) – (f). This indicates confinement of partons into mesons, in qualitative agreement with exact results in Fig. 5.

In the case when $h = 0$ and the SC term $\lambda = 0$ in Fig. 10(a), in contrast, both distributions coincide, which signals a deconfined phase. The same behavior can be observed when we include the SC term $\lambda$ for fillings $0.2 \lesssim n \lesssim 0.8$, which can be seen in Fig. 10(b) for $n = 0.248$, which again agrees with the exact results in Fig. 5. For lower fillings we observe in Fig. 10(c) that

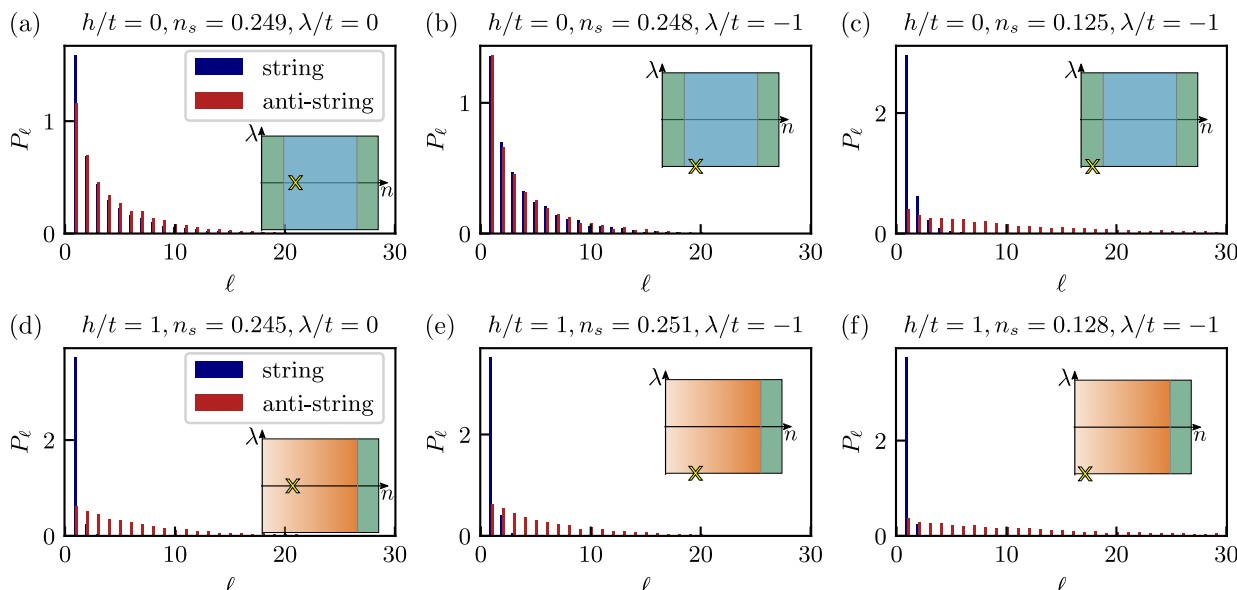

FIG. 10. String and anti-string length distributions for the mean-field theory (in the gauge sector). (a) In the absence of the $\mathbb{Z}_2$ electric field term, $h = 0$, and SC term, $\lambda = 0$, at filling $n = 0.249$, string and anti-string length distributions look similar with peaks of approximately same height, which indicates the deconfined phase. (b) By including the SC term $\lambda/t = 1$ and keeping the electric field term zero, $h = 0$, we also observe equal distributions for approximately same filling as in (a) $n = 0.248$, which signals the deconfined phase observed already in the Green's function calculations for fillings $0.2 \lesssim n \lesssim 0.8$. (c) At lower filling $n = 0.125$ for the same parameter regime as in (b) we observe that the peak at $\ell = 1$ for strings is significantly higher than the anti-string peak. Furthermore the anti-string length distribution has a significantly longer tail. Both of these observations suggest that the symmetry-broken phases for low and high fillings at $h = 0$ and $\lambda \neq$ are confining. (d) When the electric field term is non-zero, $h/t = 1$, but the SC term is zero, $\lambda = 0$, we obtain high peaks in the string length histograms for strings and low peaks with long tails for anti-strings which reflects the confined phase, for filling $n = 0.245$. (e) When also the SC term is non-zero, $\lambda/t = -1$, in addition to the electric field term, $h/t = 1$, we obtain a qualitatively similar distribution as in (d), for approximately the same filling $n = 0.251$. (f) For the same parameter as in (e) but at lower filling $n = 0.125$ we obtain the same qualitative results, meaning that there is no transition at low fillings when the $\mathbb{Z}_2$ electric field term is non-zero. The yellow "x" in all insets indicates the parameter regime in the corresponding mean-field phase diagram.

the string length peaks are higher thus indicating confinement, which is again in agreement with the Green's function results in Fig. 9.

We can conclude that both the Green's function analysis and the string and anti-string length distributions show the correct qualitative picture of confinement already on the mean-field level.

### D. Summary of the mean-field theory results

We sketch the resulting phase diagram of the mean-field theory in Fig. 11. For the $h = 0$ case, we obtain a disordered state at intermediate fillings $0.15 \lesssim n \lesssim 0.85$ which corresponds to the deconfined SPT phase in the LGT picture. For high and low fillings we also obtain transitions to confined, symmetry broken FM and AFM states, respectively. When the $\mathbb{Z}_2$ electric field term is non-zero, $h \neq 0$, we obtain a vast region of the disordered state where the Green's function results and the string-length histograms show confinement. This state thus corresponds to the confined Higgs phase in the LGT picture. In addition, for $h \neq 0$, we also observe the con-

fined AFM state for high fillings $n \gtrsim 0.85$.

The phase diagram of the mean-field model in the gauge sector Eq. (11) thus qualitatively resembles the exact model with two exceptions. Firstly, there is no parton LL or meson LL, since the U(1) symmetry is explicitly broken. Secondly, the transitions to the symmetry-broken states do not exhibit any dependence on the filling.

The mean-field theory in the charge sector can be mapped exactly to the $\mathbb{Z}_2$ LGT Eq. (1) when the electric field term is zero, $h = 0$, since one can eliminate the gauge fields, see Appendix A. The phase diagrams are thus exactly the same for $h = 0$. However, when the $\mathbb{Z}_2$ electric field term is non-zero, $h \neq 0$, the mean-field theory in the charge sector does not change since the electric term only contributes to the constant energy offset. Hence, it remains the same to the deconfined case and does not capture the confined phases we find in the full $\mathbb{Z}_2$ LGT.

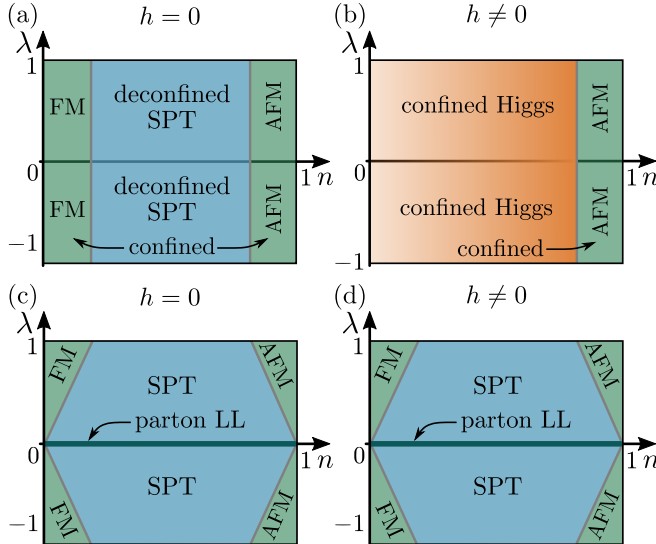

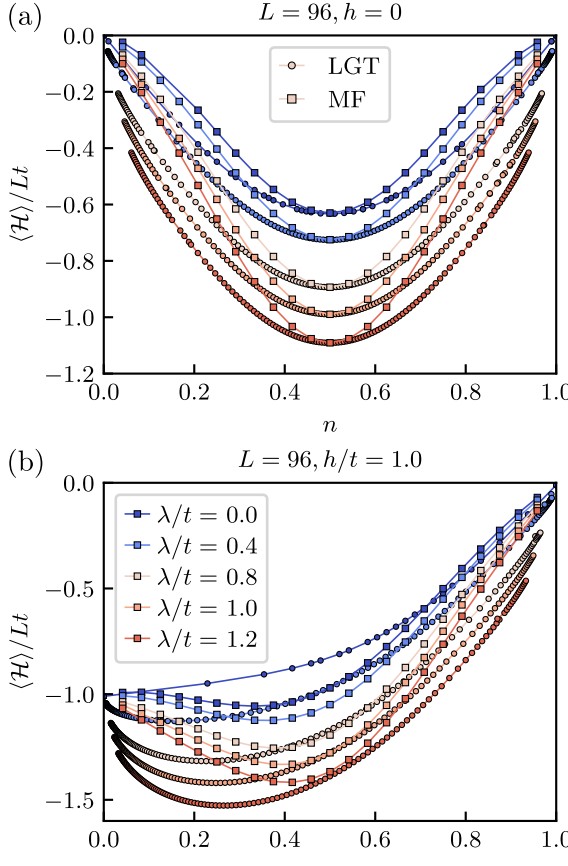

FIG. 11. A sketch of a phase diagram of the mean-field theory of the 1D $\mathbb{Z}_2$ LGT. (a) The mean-field theory in the gauge sector, Eq. (11), for $h = 0$, exhibits a disordered state for intermediate fillings $0.15 \lesssim n \lesssim 0.85$, which corresponds to a deconfined SPT state in the LGT language. In addition, we have transitions to symmetry-broken FM and AFM phases at high and low fillings. (b) At finite $\mathbb{Z}_2$ electric field term $h \neq 0$, the mean-field theory in the gauge field sector exhibits a symmetry broken AFM state for high fillings $n \gtrsim 0.85$, and a disordered state which corresponds to a confined Higgs phase. (c) The phase diagram of the mean-field theory in the matter sector for $h = 0$ is exactly the same as the phase diagram of the $\mathbb{Z}_2$ LGT for $h = 0$, where the $\mathbb{Z}_2$ fields can be eliminated and the model reduces to the superconducting model Eq. (8). (d) For $h \neq 0$, the phase diagram of the mean-field theory in the matter sector is exactly the same to the $h = 0$ case since the $\mathbb{Z}_2$ electric field term becomes a constant energy offset in the mean-field Hamiltonian in Eq. (8). In this last case the mean-field theory in the matter sector does not correctly reproduce the exact LGT results.

## V. COMPARISON BETWEEN THE MEAN-FIELD THEORY AND THE EXACT LGT

### A. Ground state energy comparison

In order to directly compare to the mean-field theory we calculate the ground state energies of the exact $\mathbb{Z}_2$ LGT Hamiltonian (1) and the mean-field Hamiltonian (11) for different values of the $\mathbb{Z}_2$ electric fields in Fig. 12. We use DMRG [39, 40] where we subtracted the chemical potential contribution from the overall ground state energy, see Appendix D 3. We observe very good qualitative agreement between both Hamiltonians. To be more precise we observe the same change of the typical free-parton parabola for $h = 0$, to a deformed concave curve for $h/t = 1$. This means that the effective mean-field theory captures the main features of the ground state energy of the exact $\mathbb{Z}_2$ LGT.

FIG. 12. Comparison of the ground state energy of the $\mathbb{Z}_2$ LGT Hamiltonian (1) and the mean-field Hamiltonian (11) for different values of the pairing term $\lambda$ as a function of filling $n$. (a) The ground state energy is symmetric around half-filling where it has a minimum in the deconfined phase $h = 0$, which is captured well by the mean-field model. (b) In the confined phase $h/t = 1$ the ground state energy rises monotonically for $\lambda = 0$ with filling $n$. A minimum reappears at finite filling when $\lambda \neq 0$. Mean-field theory matches the ground state energies well for higher fillings in the confined state. Smaller circles represent the DMRG results of the exact LGT and slightly larger squares represent the mean-field results. We only present $\lambda \geq 0$ as the energy is invariant for $\lambda \rightarrow -\lambda$.

### B. Electric polarization comparison

In order to better understand the mean-field theory we also consider the polarization of the $\mathbb{Z}_2$ electric field. We define the polarization in the $x$−direction as

$$P = \frac{1}{L+1} \sum_j \left\langle \hat{\tau}_j^x \right\rangle. \tag{17}$$

Finite value of the $\mathbb{Z}_2$ electric field polarization thus signals the FM phase. We observe that the onset of finite polarization as a function of filling $n$ changes with $\lambda$ in the case when $h = 0$, see Fig. 13(a). For $|\lambda| > 0$ the finite polarization persists for higher values of filling $n$ than for the U(1) conserving case when $\lambda = 0$, where no

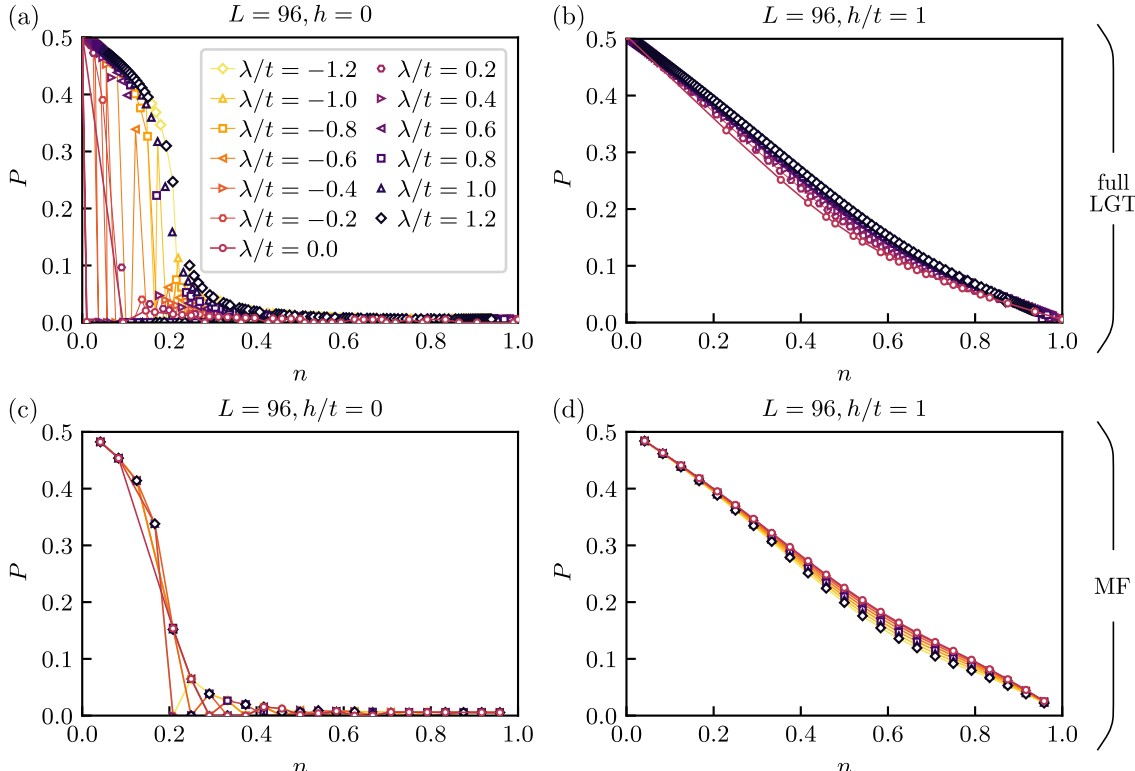

FIG. 13. Polarization of the $\mathbb{Z}_2$ electric field as a function of filling $n$ for different values of $\lambda$. (a) Exact LGT, Eq. (1), result for no $\mathbb{Z}_2$ electric field term, $h = 0$, show a non-zero polarization corresponding to the FM phase for $\lambda \neq 0$. (b) For the finite electric field term $h/t = 1$ in the exact LGT Eq. (1) we observe a monotonic decrease of the polarization with filling for every $\lambda$. (c) Mean-field theory Eq. (11) results for $h = 0$ show no dependence of polarization on $\lambda$. The curves have a similar shape to the exact LGT results in (a), with the polarization onset at a similar filling of $n \approx 0.2$. (d) Mean-field theory Eq. (11) result for $h = 1$ show the same qualitative behavior as the exact LGT results.

spontaneous polarization is found. Comparing this result directly with the mean-field theory in Fig. 13(c) we see that there is no change in polarization as a function of filling $n$ in the latter when we tune the SC term $\lambda$. The drop of polarization to zero occurs at the same value as for $\lambda \neq 0$ in the exact case. This again shows that the mean-field theory correctly captures the qualitative features of the full $\mathbb{Z}_2$ LGT, especially when $\lambda \neq 0$, when it starts to match quantitatively.

Even better agreement can be seen for the $h/t = 1$ case presented in Fig. 13(b) and Fig. 13(d), where the curves from the mean-field theory and the full $\mathbb{Z}_2$ LGT almost exactly coincide.

Finite value of polarization can also be directly related to confinement, where the string and anti-string lengths become significantly different, which results in a net polarization of the $\mathbb{Z}_2$ fields.

## VI. SUMMARY

In this work we develop a mean-field theory for a 1+1D $\mathbb{Z}_2$ LGT with a superconducting term which breaks the U(1) symmetry associated with parton number conserva-

tion. We first study the phase diagram of the $\mathbb{Z}_2$ LGT as a function of the SC term $\lambda$ and filling $n$ at different values of the $\mathbb{Z}_2$ electric field $h$, which we sketch in Fig. 1(c) and (d). We use DMRG and analytical mappings to known models, like the Kitaev chain and the transverse field Ising model.

The $\mathbb{Z}_2$ LGT reduces to a free parton model in the absence of the electric field term, $h = 0$, and maps to a Kitaev chain, which is equivalent to a transverse field Ising model. The system forms a free parton LL when the $\mathbb{Z}_2$ electric field term and the SC term are both zero and the U(1) symmetry in the charges is conserved. The Kitaev chain that the $\mathbb{Z}_2$ LGT maps to at $h = 0$ is known to host SPT phases when $\lambda \neq 0$ at intermediate fillings $0.15 \lesssim n \lesssim 0.85$; for lower and higher fillings it undergoes a transition to symmetry broken states. These are topologically trivial and map to FM and AFM phases of the $\mathbb{Z}_2$ electric fields. In our analysis we pointed out that the symmetry-broken phases should be viewed as confined phases, featuring mesonic pairs of partons arising from pair-fluctuations generated by the $\lambda$-term in the Hamiltonian.

When the $\mathbb{Z}_2$ electric field term is non-zero, $h \neq 0$, the partons always confine into mesons in $1 + 1$D. For $\lambda = 0$

mesons form a LL which is protected by the U(1) symmetry. For $\lambda \neq 0$ the system forms a confined Higgs phase, which undergoes a transition to a symmetry-broken AFM state for high filling $n > 0.85$.

We then derived the mean-field theory of the $\mathbb{Z}_2$ LGT, by factorizing matter and gauge degrees of freedom and enforcing the Gauss law on a mean-field level. The resulting mean-field Hamiltonian for the matter sector is a superconducting quantum wire model, where the gauge field renormalizes the interaction parameters. Such model can be solved via the Bogoliubov transformation for a given set of parameters and filling.

The obtained mean-field model in the $\mathbb{Z}_2$ gauge-field sector is an Ising model with transverse and longitudinal fields, which we solve self-consistently for a given set of parameter values and fillings. The value of the Ising interaction is directly related to the chemical potential, the longitudinal field is directly proportional to the $\mathbb{Z}_2$ electric field term $h$, and the transverse field is related to the hopping amplitude $t$ and superconducting term $\lambda$ renormalized by the matter field. In order to solve the mean-field theory of the $\mathbb{Z}_2$ fields we first determine the value of the longitudinal field, by solving the corresponding matter mean-field Hamiltonian via the Bogoliubov transformation. Next, we find the correct value for the Ising interaction, for the given set of parameters and fillings which yields the correct filling $n$. We do this by running the DMRG for different values of $\mu_\tau$, until we find the value which reduces the difference between the target and actually obtained filling.

We studied the phase diagram of the mean-field theory and compared it to the $\mathbb{Z}_2$ LGT results, which we outlined above. We found overall remarkable qualitative agreement between the $\mathbb{Z}_2$ LGT results and the mean-field theory. The mean-field theory correctly captures the confined and deconfined phases, and the transitions to the symmetry-breaking states. The lack of the U(1) symmetry results in the absence of the Luttinger liquid phases, however the confined and deconfined phases are still qualitatively captured.

Our work thus shows that a complicated $\mathbb{Z}_2$ LGT has a simple mean-field theory description, which captures all of the important features of the LGT. We believe that the one-dimensional mean-field theory can be extended to higher dimensions and could offer new insights for higher-dimensional $\mathbb{Z}_2$ LGTs whose phase diagrams are still not fully established.

# VII. ACKNOWLEDGEMENTS

We thank Monika Aidelsburger, Luca Barbiero, Lukas Homeier, Hannah Lange, Simon Linsel, and Felix Palm for fruitful discussions. Our work was funded by the Deutsche Forschungsgemeinschaft (DFG, German Research Foundation) under Germany's Excellence Strategy – EXC-2111 – 390814868 and via Research Unit FOR 2414 under project number 277974659, and received funding from the European Research Council (ERC) under the European Union's Horizon 2020 research and innovation programm (Grant Agreement no 948141) — ERC Starting Grant SimUcQuam. F.G. acknowledge funding within the QuantERA II Programme that has received funding from the European Union's Horizon 2020 research and innovation programme under Grand Agreement No 101017733, support by the QuantERA grant DYNAMITE and by the Deutsche Forschungsgemeinschaft (DFG, German Research Foundation) under project number 499183856.

# Appendix A: Mapping between the $\mathbb{Z}_2$ LGT and the Kitaev chain

## 1. Mapping of the $\mathbb{Z}_2$ LGT at $h = 0$ to the 1D superconducting model

We start by considering the $\mathbb{Z}_2$ lattice gauge theory from the main text Eq. (1) in the regime when $h = 0$

$$\hat{\mathcal{H}} = -t \sum_{\langle i,j \rangle} \left( \hat{a}_i^\dagger \hat{\tau}_{\langle i,j \rangle}^z \hat{a}_j + \text{h.c.} \right)$$
$$+ \lambda \sum_{\langle i,j \rangle} \left( \hat{a}_i^\dagger \hat{\tau}_{\langle i,j \rangle}^z \hat{a}_j^\dagger + \text{h.c.} \right) + \mu \sum_j \hat{n}_j. \quad \text{(A1)}$$

In order to introduce or eliminate the $\mathbb{Z}_2$ fields one can attach a string of $\mathbb{Z}_2$ gauge fields to the hard-core matter operators and construct dressed partons which can be written as [4, 24, 28]

$$\hat{b}_j^\dagger = \left( \prod_{l<j} \hat{\tau}_{l,l+1}^z \right) \hat{a}_j^\dagger, \quad \hat{b}_j = \left( \prod_{l<j} \hat{\tau}_{l,l+1}^z \right) \hat{a}_j, \quad \text{(A2)}$$

where, $\hat{b}_j^\dagger$ ($\hat{b}_j$) is a dressed hard-core boson creation (annihilation) operator and $\hat{\tau}_{j,j+1}^z$ is the Pauli matrix in the $z$-basis, representing the $\mathbb{Z}_2$ gauge field on the link. We note that charges and $\mathbb{Z}_2$ fields commute.

Such construction follows from the Gauss law constraint Eq. (2), where we set $G_j = +1, \forall j$. Hence, a charge creation or annihilation has to flip the configuration of the $\mathbb{Z}_2$ electric field which are represented with the Pauli matrices in the $x$-basis, $\hat{\tau}_{j,j+1}^x$. The $\mathbb{Z}_2$ strings can be attached either from the left or from the right and run until the lattice end. In Eq. (A2) we defined them from the left, however both definitions work.

By applying the product of string operators on both sides of equalities in Eq. (A2) we can express the hard-core operators in Eq. (A1) in terms of the new dressed operators as

$$\hat{a}_j^\dagger = \left( \prod_{l<j} \hat{\tau}_{l,l+1}^z \right) \hat{b}_j^\dagger, \quad \hat{a}_j = \left( \prod_{l<j} \hat{\tau}_{l,l+1}^z \right) \hat{b}_j. \quad \text{(A3)}$$

We can use the above expressions and replace the hard-core boson operators in Eq. (A1) with the dressed hard-

core operators which eliminates the $\mathbb{Z}_2$ gauge fields

$$\hat{\mathcal{H}} = -t \sum_j \left( \hat{b}_j^\dagger \hat{b}_{j+1} + \text{h.c.} \right)$$
$$+ \lambda \sum_j \left( \hat{b}_{j+1}^\dagger \hat{b}_j^\dagger + \text{h.c.} \right) + \mu \sum_j \left( \hat{b}_j^\dagger \hat{b}_j - 1/2 \right). \quad \text{(A4)}$$

This expression already has the same form as the 1D superconducting model, studied by Kitaev [35, 36]. We note that the operators in Eq. (A4) are hard-core bosons and not fermions as is the case in the 1D superconducting model.

However, in one dimension, one can use the Jordan-Wigner transformation [55, 56] and map the above hard-core bosonic model in Eq. (A4) to the fermionic 1D superconducting model. We attach the so called Jordan-Wigner strings to hard-core bosons in order to obtain spinless fermionic operators [50, 55, 56]

$$\hat{c}_j^\dagger = \left( \prod_{l<j} e^{i\pi \hat{n}_l} \right) \hat{b}_j^\dagger, \quad \hat{c}_j = \left( \prod_{l<j} e^{-i\pi \hat{n}_l} \right) \hat{b}_j. \quad \text{(A5)}$$

Here $\hat{c}^\dagger$ ($\hat{c}$) is the new fermionic creation (annihilation) operator. By applying the string operators to the both sides of the equations in Eq. (A5) we can express the dressed hard-core operators with the fermionic operators

$$\hat{b}_j^\dagger = \left( \prod_{l<j} e^{i\pi \hat{n}_l} \right) \hat{c}_j^\dagger, \quad \hat{b}_j = \left( \prod_{l<j} e^{-i\pi \hat{n}_l} \right) \hat{c}_j. \quad \text{(A6)}$$

By using Eq. (A6) and replacing the dressed hard-core operators with the newly defined spinless fermion operators in model (A4) we obtain the 1D superconducting model, studied by Kitaev [35, 36]

$$\hat{\mathcal{H}}_K = -t \sum_j \left( \hat{c}_j^\dagger \hat{c}_{j+1} + \text{h.c.} \right)$$
$$- \lambda \sum_j \left( \hat{c}_{j+1}^\dagger \hat{c}_j^\dagger + \text{h.c.} \right) + \mu \sum_j \left( \hat{c}_j^\dagger \hat{c}_j - 1/2 \right). \quad \text{(A7)}$$

Here the sign in front of $\lambda$ changed due to the Jordan-Wigner string, and is different from the usual formulation of the 1D superconducting chain [35, 36]. We thus showed that the full $\mathbb{Z}_2$ lattice gauge theory Eq. (1) without the $\mathbb{Z}_2$ electric field term directly maps to the Kitaev chain when $\lambda = -t$ as was already shown in Ref. [28].

We would like to note that by a complex unitary transformation of the charge operators

$$\hat{a}_j^\dagger \to i \hat{a}_j^\dagger, \quad \hat{a}_j \to -i \hat{a}_j, \quad \text{(A8)}$$

the sign of the superconducting term changes and the full Hamiltonian (1) from the main text is now equal to

$$\hat{\mathcal{H}} = -t \sum_{\langle i,j \rangle} \left( \hat{a}_i^\dagger \hat{\tau}_{\langle i,j \rangle}^z \hat{a}_j + \text{h.c.} \right) - h \sum_{\langle i,j \rangle} \hat{\tau}_{\langle i,j \rangle}^x$$
$$- \lambda \sum_{\langle i,j \rangle} \left( \hat{a}_i^\dagger \hat{\tau}_{\langle i,j \rangle}^z \hat{a}_j^\dagger + \text{h.c.} \right) + \mu \sum_j \hat{n}_j. \quad \text{(A9)}$$

Borla et al. in Ref. [28] argued that both regimes $\lambda > 0$ and $\lambda < 0$ form a nontrivial symmetry protected state, where the SC term $\lambda$ opens a gap. Furthermore, these two phases are distinct since the time-reversal symmetry was not preserved with the mapping in Eq. (A8).

In our calculations we observe that the physical observable are generally invariant to the transformation Eq. (A8), as they remain identical for $\lambda \to -\lambda$. The only difference are the entanglement entropy calculations where the $S(L/2)$ is qualitatively much larger for $\lambda > 0$ than $\lambda < 0$, see Fig. 2.

## 2. Including the $\mathbb{Z}_2$ electric field term $h$

We note that eliminating the $\mathbb{Z}_2$ electric field term by attaching the $\mathbb{Z}_2$ strings results in highly non-local $h\hat{\tau}^x$-term in the Hamiltonian [24]

$$\hat{\tau}_{j,j+1}^x = e^{i\pi \sum_{l<j} \hat{n}_l}. \quad \text{(A10)}$$

The above expression is obtained by considering open boundary conditions where we assume that the chain starts and ends with an anti-string ($\tau^x = +1$). By fixing the Gauss law to the physical sector ($\hat{G}_j |\psi\rangle = +1 |\psi\rangle$) we can express $\mathbb{Z}_2$ electric field on site $j$ as a product of Gauss law operators

$$\prod_{l<j} \hat{G}_l = (-1)^{\sum_{l<j} \hat{n}_j} \hat{\tau}_{\langle j,j+1 \rangle}^x = \mathbb{I}. \quad \text{(A11)}$$

Applying $\hat{\tau}_{\langle j,j+1 \rangle}^x$ on both side of the above equation gives us the Eq. (A10) [24].

## 3. Mapping between the Kitaev chain and the transverse-field Ising model

As mentioned in the main text, the Kitaev chain can be formally mapped to the transverse field Ising chain, when $\lambda = -t$. This can be done by using the Jordan-Winger transformation which yields [36]

$$\hat{\mathcal{H}}_I = -J \sum_j \hat{\sigma}_j^x \hat{\sigma}_{j+1}^x - h_z \sum_j \hat{\sigma}_j^z, \quad \text{(A12)}$$

where $J = t$ and $h_z = -\frac{\mu}{2}$. Here Kitaev and Laumann in Ref. [36] consider empty or vacant fermion sites as spin-$1/2$ up and down configurations in the $z$-basis, respectively. Hence spins $\hat{\sigma}$ are not directly connected to the $\mathbb{Z}_2$ fields, where anti aligned $\mathbb{Z}_2$ electric fields on the neighboring links signal a presence of a particle on the lattice site, i.e., spins $\sigma$ are defined on the lattice sites, whereas the $\mathbb{Z}_2$ fields $\hat{\tau}$ reside on lattice links. However, by performing the duality transformation

$$\hat{\sigma}_j^z \to \hat{\tau}_{j-1,j}^x \hat{\tau}_{j,j+1}^x, \quad \hat{\sigma}_j^x \hat{\sigma}_{j+1}^x \to \hat{\tau}_{j,j+1}^z, \quad \text{(A13)}$$

one can rewrite the model in Eq. (A12) as

$$\hat{\mathcal{H}}_I = -J \sum_j \hat{\tau}^z_{j,j+1} - h_z \sum_j \hat{\tau}^x_{j-1,j}\hat{\tau}^x_{j,j+1}. \qquad (A14)$$

This is exactly the same Hamiltonian as the spin-1/2 model of the full $\mathbb{Z}_2$ LGT Eq. (1) for $\lambda = -t$ and $h = 0$, after integrating out the charges via the Gauss law constraint, discussed in the next section.

### Appendix B: Mapping of the $\mathbb{Z}_2$ LGT to the spin-1/2 model

We use the Gauss law Eq. (2) to define the local number operator, by choosing the so called physical sector without background charges [4]

$$\hat{G}_j = \hat{\tau}^x_{\langle j-1,j \rangle}\hat{\tau}^x_{\langle j,j+1 \rangle}(-1)^{\hat{n}_j} = +\mathbb{I}, \quad \forall j. \qquad (B1)$$

This allows us to express the local density operator in terms of the $\mathbb{Z}_2$ electric strings

$$\hat{n}_j = \frac{1}{2}\left(1 - \hat{\tau}^x_{j-1,j}\hat{\tau}^x_{j,j+1}\right). \qquad (B2)$$

As a result a domain wall in the $\mathbb{Z}_2$ electric fields signals a presence of a particle on the physical lattice site.

Using the above result we can rewrite the charge creation and annihilation operators in terms of the $\mathbb{Z}_2$ electric and gauge fields as

$$\hat{a}^\dagger_j = \left(\prod_{l<j} \hat{\tau}^z_{l,l+1}\right)\frac{1}{2}\left(1 + \hat{\tau}^x_{j-1,j}\hat{\tau}^x_{j,j+1}\right),$$
$$\hat{a}_j = \left(\prod_{l<j} \hat{\tau}^z_{l,l+1}\right)\frac{1}{2}\left(1 - \hat{\tau}^x_{j-1,j}\hat{\tau}^x_{j,j+1}\right). \qquad (B3)$$

We once again note that we always consider that our chain starts and ends with a link. The logic of the operator is following: we first take into account that the particle at some site $j$ can be added only if the site $j$ is empty, otherwise the state has to be annihilated. This is achieved by first applying the operator $\frac{1}{2}\left(1 + \hat{\tau}^x_{j-1,j}\hat{\tau}^x_{j,j+1}\right)$ which projects to the state with an empty site $j$. After this we apply the product of $\prod_{l<j} \hat{\tau}^z_{l,l+1}$ operators up to the link which attaches to our site from the left. In such way we create a domain wall and thus a particle at that site. The same logic applies for the annihilation operator.

Such mapping is reminiscent of the Jordan-Wigner mapping used by Kitaev and Laumann in Ref. [36] and to the string attachment used by Borla in Ref. [24] to translate the $\mathbb{Z}_2$ LGT to free fermions for $h = 0$. However the charge operators are here written exclusively in the $\mathbb{Z}_2$ spin language where we took into account the Gauss law and we do not use any Majorana operators.

Using mapping Eq. (B3) together with Eq. (B2) we can rewrite the hopping term and the $\mathbb{Z}_2$ electric field term in Hamiltonian Eq. (1) from the main text purely in the spin-language as [24–28]

$$\hat{\mathcal{H}}^s_{LGT} = t\sum_{j=2}^{L-1}\left(4\hat{S}^x_{j-1}\hat{S}^x_{j+1}\hat{S}^z_j - \hat{S}^z_j\right) - h\sum_{j=1}^{L} 2\hat{S}^x_j, \quad (B4)$$

where we replaced the Pauli matrices with spin-1/2 operators: $\hat{\tau}^{x,z}_{\langle i,i+1 \rangle} = 2\hat{S}^{x,z}_j$. Similarly, we can also express the superconducting term as

$$\hat{\mathcal{H}}^s_\lambda = \lambda\sum_{j=2}^{L-1}\left(4\hat{S}^x_{j-1}\hat{S}^x_{j+1}\hat{S}^z_j + \hat{S}^z_j\right) \qquad (B5)$$

and a term which is proportional to the chemical potential $\hat{\mathcal{H}}^s_\mu = \mu \sum_{j=1}^{L-1} 2\hat{S}^x_j\hat{S}^x_{j+1}$. This term is important for the DMRG simulations, since it is used to control the filling in the lattice.

We can consider two different limits $\lambda = \pm t$, where the Hamiltonian simplifies. When $\lambda = -t$ we get an Ising model with transverse and longitudinal field

$$\hat{\mathcal{H}}^s = -t\sum_{j=2}^{L-1} 2\hat{S}^z_j - h\sum_{j=1}^{L} 2\hat{S}^x_j + \mu\sum_{j=1}^{L-1} 2\hat{S}^x_j\hat{S}^x_{j+1}. \quad (B6)$$

Such model is similar to the mean-field Hamiltonian for the gauge fields Eq. (11). Furthermore, we note that this Hamiltonian maps exactly to the transverse field Ising model which one obtains after applying the Jordan-Wigner transformation to the Kitaev chain, discussed in Appendix A 3.

In the other limit when $\lambda = t$, we get a slightly different but equivalent model

$$\hat{\mathcal{H}}^s = t\sum_{j=2}^{L-1} 8\hat{S}^x_{j-1}\hat{S}^x_{j+1}\hat{S}^z_j - h\sum_{j=1}^{L} 2\hat{S}^x_j + \mu\sum_{j=1}^{L-1} 2\hat{S}^x_j\hat{S}^x_{j+1}. \qquad (B7)$$

The ungauged superconducting model Eq. (A7) is symmetric in $\lambda \to -\lambda$, hence the results should be the same for $\lambda > 0$ and $\lambda < 0$. However, as we have just demonstrated, the mapping to the spin Hamiltonian does not yield a Hamiltonian which posses this symmetry. Although the Hamiltonians are unitary equivalent, we believe that DMRG calculations for $\lambda > 0$ are more complicated and thus yield higher entanglement entropy than for the $\lambda < 0$ regime, since they require higher bond dimension. Moreover if the time-reversal symmetry is considered, a simple transformation $\lambda \to -\lambda$ does not yield the same Hamiltonian, which indicates that these are distinct phases as argued in [28].

## Appendix C: Details on the derivation of the mean-field theory

### 1. Derivation of the mean-field models

We start our derivation by assuming the following ansatz already presented in the main text

$$|\psi\rangle = |\psi_\tau\rangle \otimes |\psi_c\rangle, \tag{C1}$$

$$\hat{\mathcal{H}} = -t \sum_{j=1}^{L-1} \left( \langle \hat{c}_{j+1}^\dagger \hat{c}_j \rangle + \langle \hat{c}_j^\dagger \hat{c}_{j+1} \rangle \right) \hat{\tau}_{\langle j,j+1 \rangle}^z + \lambda \sum_{j=1}^{L-1} \left( \langle \hat{c}_{j+1}^\dagger \hat{c}_j^\dagger \rangle + \langle \hat{c}_j \hat{c}_{j+1} \rangle \right) \hat{\tau}_{\langle j,j+1 \rangle}^z - h \sum_{j=0}^{L} \hat{\tau}_{\langle j,j+1 \rangle}^x$$
$$-t \sum_{j=1}^{L-1} \left( \langle \hat{\tau}_{\langle j,j+1 \rangle}^z \rangle \hat{c}_{j+1}^\dagger \hat{c}_j + \text{h.c.} \right) + \lambda \sum_{j=1}^{L-1} \left( \langle \hat{\tau}_{\langle j,j+1 \rangle}^z \rangle \hat{c}_{j+1}^\dagger \hat{c}_j^\dagger + \text{h.c.} \right), \tag{C2}$$

where operators within the angle brackets are to be considered as the average expectation values, i.e., their values on the mean-field level. We also switched to the fermionic matter, $\hat{c}$, which is an equivalent description in one-dimension due to the Jordan-Wigner transformation (as demonstrated in Section A). This simplifies our calculations as we can use the standard Bogoliubov transformation in the next steps. In addition, we dropped the constant offsets and the chemical potential term.

In the above expression we effectively decouple the charges from the $\mathbb{Z}_2$ gauge fields. We can thus write two separate Hamiltonians for the charges and the $\mathbb{Z}_2$ fields, where we re-introduce the chemical potential in forms of Lagrange multipliers. These enforce the desired filling and the Gauss law on the mean-field level. The resulting mean-field model for the charges from Eq. (C2) can be written as

$$\hat{\mathcal{H}}_c = -t_c \sum_{j=1}^{L-1} \left( \hat{c}_{j+1}^\dagger \hat{c}_j + \text{h.c.} \right)$$
$$+ \lambda_c \sum_{j=1}^{L-1} \left( \hat{c}_{j+1}^\dagger \hat{c}_j^\dagger + \text{h.c.} \right) + \mu_c \sum_{j=1}^{L} \left( \hat{n}_j - n \right), \tag{C3}$$

where we defined $t_c = t\langle \hat{\tau}_{\langle j,j+1 \rangle}^z \rangle$ and $\lambda_c = \lambda \langle \hat{\tau}_{\langle j,j+1 \rangle}^z \rangle$, and added the chemical potential term $\mu_c$ which enforces the desired filling of the chain.

Moreover we obtain the mean-field model for the $\mathbb{Z}_2$ gauge fields as

$$\hat{\mathcal{H}}_\tau = -g \sum_{j=1}^{L-1} \hat{\tau}_{\langle j,j+1 \rangle}^z - h \sum_{j=0}^{L} \hat{\tau}_{\langle j,j+1 \rangle}^x$$
$$+ \mu_\tau \sum_{j=1}^{L} \left( \hat{\tau}_{j-1,j}^x \hat{\tau}_{j,j+1}^x - (1 - 2n) \right), \tag{C4}$$

where we factorize matter and gauge field. This gives us the following model Hamiltonian,

where we defined

$$g = t \left( \left\langle \hat{c}_{j+1}^\dagger \hat{c}_j \right\rangle + \left\langle \hat{c}_j^\dagger \hat{c}_{j+1} \right\rangle \right)$$
$$- \lambda \left( \left\langle \hat{c}_{j+1}^\dagger \hat{c}_j^\dagger \right\rangle + \left\langle \hat{c}_j \hat{c}_{j+1} \right\rangle \right), \tag{C5}$$

as in the main text. In addition we added the Lagrange multiplier $\mu_\tau$ which enforces the Gauss law constraint and the correct effective filling. It can be thus considered as a chemical potential term. This term is obtained directly from the Gauss law, when we consider the physical sector $G_i = +1, \forall i$ which yields the relation Eq. (B2), explained in Appendix B. Summing the Eq. (B2) over all lattice sites yields

$$Ln = \frac{1}{2} \left( L - \sum_{j=1}^{L} \hat{\tau}_{\langle j-1,j \rangle}^x \hat{\tau}_{\langle j,j+1 \rangle}^x \right), \tag{C6}$$

where $L$ is the length of the chain and $n$ is the average density of the particles. By rearranging Eq. (C6) we obtain Eq. (10) in the main text.

### 2. Solving the mean-field theory for the charges

The mean-field Hamiltonian for the charges Eq. (C3) is in fact a superconducting quantum wire model which can be solved using the Bogoliubov transformation [35, 38]. We can first perform the Fourier transformation $\hat{c}_j^\dagger = \frac{1}{\sqrt{L}} \sum_k e^{-ikj} \hat{c}_k^\dagger$ and rewrite the Hamiltonian in Eq. (C3) as

$$\hat{\mathcal{H}}_c = \sum_{k>0} \left( \mu_c - 2t_c \cos(k) \right) \left( \hat{c}_k^\dagger \hat{c}_k + \hat{c}_{-k}^\dagger \hat{c}_{-k} \right) \tag{C7}$$
$$+ \sum_{k>0} 2i\lambda_c \sin(k) \left( -\hat{c}_k^\dagger \hat{c}_{-k}^\dagger + \hat{c}_{-k} \hat{c}_k \right) \tag{C8}$$
$$- \mu_c n L, \tag{C9}$$

where we only consider the positive Fourier modes $k$. In order to diagonalize the expression Eq. (C9) we use the Bogoliubov transformation and write [38]

$$
\hat{\mathcal{H}}_c = \sum_{k>0} \begin{pmatrix} \hat{c}_k^\dagger & \hat{c}_{-k} \end{pmatrix} \begin{pmatrix} \epsilon(k) & -2i\lambda_c \sin(k) \\ 2i\lambda_c \sin(k) & -\epsilon(k) \end{pmatrix} \begin{pmatrix} \hat{c}_k \\ \hat{c}_{-k}^\dagger \end{pmatrix} \\ + \sum_{k>0} \epsilon(k) - \mu_c n L,
$$
(C10)

where we wrote $\epsilon(k) = \mu_c - 2t_c \cos(k)$. This can be diagonalized as

$$
\hat{\mathcal{H}}_c - \sum_{k>0} \epsilon(k) + \mu_c n L = \sum_{k>0} \begin{pmatrix} \hat{c}_k^\dagger & \hat{c}_{-k} \end{pmatrix} \boldsymbol{H}(k) \begin{pmatrix} \hat{c}_k \\ \hat{c}_{-k}^\dagger \end{pmatrix} \\ = \sum_{k>0} \begin{pmatrix} \hat{b}_k^\dagger & \hat{b}_{-k} \end{pmatrix} \boldsymbol{\Lambda}(k) \begin{pmatrix} \hat{b}_k \\ \hat{b}_{-k}^\dagger \end{pmatrix}, \quad \text{(C11)}
$$

where $\boldsymbol{\Lambda}$ is the diagonal matrix with entries

$$
\Lambda_\pm(k) = \pm\sqrt{(\mu_c - 2t_c \cos(k))^2 + (2\lambda_c \sin(k))^2}. \quad \text{(C12)}
$$

The operators $\hat{b}^{(\dagger)}$ are linear combination of the operators $\hat{c}$ and $\hat{c}^\dagger$ related by the Bogoliubov transformation [38]. The resulting diagonalized Hamiltonian can be written as

$$
\hat{\mathcal{H}}_c = \sum_k \Lambda_+ \hat{b}_k^\dagger \hat{b}_k - \sum_{k>0} \Lambda_+ + \sum_{k>0} \epsilon(k) - \mu_c n L. \quad \text{(C13)}
$$

The ground state energy per lattice site is thus equal to

$$
E_0 = \left\langle \hat{\mathcal{H}}_c \right\rangle / L = -\frac{1}{L} \sum_{k>0} \Lambda_+ + \frac{1}{L} \sum_{k>0} \epsilon(k) - \mu_c n, \quad \text{(C14)}
$$

which in the thermodynamical limit equals to

$$
E_0 = -\frac{1}{2\pi} \int_0^\pi \mathrm{d}k \sqrt{(\mu_c - 2t_c \cos(k))^2 + (2\lambda_c \sin(k))^2} \\ + \frac{1}{2\pi} \int_0^\pi \mathrm{d}k \, (\mu_c - 2t_c \cos(k)) - \mu_c n.
$$
(C15)

For a given chemical potential $\mu_c$ we can thus find the ground state energy. In order to connect the chemical potential to the correct filling we need to solve the self-consistency equation

$$
0 \stackrel{!}{=} \frac{dE_0}{d\mu_c} = \\ = -\frac{1}{2\pi} \int_0^\pi \mathrm{d}k \frac{\epsilon(k)}{\sqrt{\epsilon^2(k) + (2\lambda_c \sin(k))^2}} + \frac{1}{2} - n.
$$
(C16)

This gives us the equation for the filling of the chain,

$$
n = \frac{1}{2} \left( 1 - \frac{1}{\pi} \int_0^\pi \mathrm{d}k \frac{\epsilon(k)}{\sqrt{\epsilon^2(k) + (2\lambda_c \sin(k))^2}} \right). \quad \text{(C17)}
$$

These integrals can be performed numerically to obtain the ground state energies by solving Eq. C17 self-consistently and finding the correct $\mu_c$ for a desired filling $n$.

We note that the above calculations can be simplified if $\lambda = 0$, as then we simply have to calculate the expression $\langle \hat{c}_{j+1}^\dagger \hat{c}_j \rangle$ for free fermions, which after some simple algebra yields $\langle \hat{c}_{j+1}^\dagger \hat{c}_j \rangle = \frac{\sin(\pi n)}{\pi}$.

### 3. Solving the mean-field theory for the gauge fields

The mean-field model for the $\mathbb{Z}_2$ fields is an Ising model with transverse and longitudinal field Eq. (C4) which can not directly be solved analytically. For that reason we use numerical simulations, where we employ DMRG. However, in order to simulate the model, we have to first determine the parameter $g$, Eq (C5), which depends on the hopping amplitude $t$, superconducting pairing term $\lambda$, and filling $n$. For that we use the result from the previous section since we realize that

$$
E_0 = -t_c \left( \langle \hat{c}_{j+1}^\dagger \hat{c}_j \rangle + \langle \hat{c}_j^\dagger \hat{c}_{j+1} \rangle \right) \\ + \lambda_c \left( \langle \hat{c}_{j+1}^\dagger \hat{c}_j^\dagger \rangle + \langle \hat{c}_j \hat{c}_{j+1} \rangle \right) = -\langle \hat{\tau}_{\langle j,j+1 \rangle}^z \rangle g.
$$

In the second equality we recognize that $E_0$ is just Eq. (C5), with renormalized $t$ and $\lambda$ by $\langle \hat{\tau}_{\langle j,j+1 \rangle}^z \rangle$, i.e., we took into account the definition of $t_c$ and $\lambda_c$. Hence, we can calculate $g$ as

$$
g = \frac{1}{2\pi} \int_0^\pi \mathrm{d}k \sqrt{(\tilde{\mu}_c - 2t \cos(k))^2 + (2\lambda \sin(k))^2} \\ + \tilde{\mu}_c \left( n - \frac{1}{2} \right). \quad \text{(C18)}
$$

where we simply normalize expression Eq. (C15) with $\langle \hat{\tau}_{\langle j,j+1 \rangle}^z \rangle$. This means that we only have to find the correct chemical potential $\tilde{\mu}_c = \mu_c / \langle \hat{\tau}_{\langle j,j+1 \rangle}^z \rangle$ which yields the correct filling for which we want to solve the mean-field model Eq. (C5). This is how we obtained the Eq. (14) from the main text, which is just the modified equation Eq. (C17), where the parameters are simply normalized by $\langle \hat{\tau}_{\langle j,j+1 \rangle}^z \rangle$

$$
n = \frac{1}{2} \left( 1 - \frac{1}{\pi} \int_0^\pi \mathrm{d}k \frac{\tilde{\mu}_c - 2t \cos(k)}{\sqrt{(\tilde{\mu}_c - 2t \cos(k))^2 + \lambda^2(k)}} \right). \quad \text{(C19)}
$$

In fact all of the above calculations can be done in terms of $\tilde{\mu}_c$ and we do not need to know the actual value of $\langle \hat{\tau}_{\langle j,j+1 \rangle}^z \rangle$.

Once we obtain the correct value for $g$ we can solve the mean-field model for the $\mathbb{Z}_2$ fields. This has to be done again self-consistently, by finding the correct $\mu_\tau$ which

yields the correct filling $n$, which is calculated from the function Eq. (B2) We do this by using DMRG and find the best value for $\mu_\tau$, which we explain in Appendix D.

## Appendix D: DMRG calculations

### 1. DMRG simulation of the $\mathbb{Z}_2$ LGT

We simulate the $\mathbb{Z}_2$ LGT Hamiltonian with DMRG, by considering the $\mathbb{Z}_2$ LGT mapped to the spin-1/2 model which we derived in Appendix B. The resulting exact Hamiltonian is [24–28]

$$\hat{\mathcal{H}}_{\mathrm{DMRG}} = t \sum_{j=2}^{L-1} \left( 4\hat{S}_{j-1}^x \hat{S}_{j+1}^x \hat{S}_j^z - \hat{S}_j^z \right)$$
$$+ \lambda \sum_{j=2}^{L-1} \left( 4\hat{S}_{j-1}^x \hat{S}_{j+1}^x \hat{S}_j^z + \hat{S}_j^z \right)$$
$$- h \sum_{j=1}^{L} 2\hat{S}_j^x + \mu \sum_{j=1}^{L-1} 2\hat{S}_j^x \hat{S}_{j+1}^x, \quad \text{(D1)}$$

where we added the chemical potential term $\mu$ in order to control the filling of our chain and we replaced the Pauli matrices with spin-1/2 operators: $\hat{\tau}_{\langle j,j+1\rangle}^{x,z} = 2\hat{S}_j^{x,z}$. Note that in order to simulate the chain with length $L$ we have to simulate a spin chain with length $L+1$.

The filling in such model is calculated via Eq. (B2) and the Green's function Eq. (6) is constructed by using the mapping in Eq. (B3).

### 2. DMRG simulation of the mean-field model

The effective mean-field theory is already a spin-1/2 Hamiltonian meaning that the DMRG calculations can be directly implemented in the spin-1/2 language by directly simulating Hamiltonian (11).

We have to fix the density $n$ and then vary the chemical potential $\mu_\tau$. The value of $g$ has to already be calculated ahead since it is specific to chosen parameter values $t$ and $\lambda$ and filling $n$, see Appendix C. The procedure to solve the mean-field theory is thus as following:

1. We chose the parameter values $t$ and $\lambda$ and a filling $n$ for which we want to solve our mean-field theory Eq. (11). We calculate the $g$ for a given $n$ by finding the optimal $\tilde{\mu}_c$, via Eq. (13) and Eq. (14).

2. We simulate the spin model Eq. (11) by using DMRG. We use $g$ which we calculated in the previous step and choose a $\mathbb{Z}_2$ electric field strength $h$. The only missing parameter now is $\mu_\tau$. To find the correct $\mu_\tau$ which corresponds to the correct filling $n$ we run the following procedure:

(a) We define $\mu_{min}$ and $\mu_{max}$ and find a ground state of the system for $\mu_{min} < \mu_{p=0} < \mu_{max}$ with DMRG and calculate the corresponding filling $n(p = 1)$. The starting value is typically chosen to be $\mu_{p=0} = 0.1t$.

(b) For the given $\mu_p$ we calculate the filling by using Eq. (B2). If the target filling $n$ is greater than the calculated filling $n(p = 1)$ we redefine $\mu_{max} = \mu_j$. In contrast if the target filling is smaller than the calculated filling we redefine $\mu_{min} = \mu_j$.

(c) We define the new chemical potential as $\mu_j = \frac{1}{2}(\mu_{min} + \mu_{max})$ and calculate the corresponding filling $n(p)$.

(d) We repeat steps b) and c) for 15 times which gives us high accuracy for $\mu_\tau$ which gives us the ground state solution with a filling close to the chosen $n$. We generally exclude the data if the error between the target filling and the filling from the last step $p_f$ is greater than one percent, $\frac{n-n(p_f)}{n} > 0.01$.

### 3. Ground state energy comparison

When comparing the ground state energies of both models the chemical potential term contribution was subtracted by subtracting the term $\mu L(1 - 2n)$ where $n$ is the filling of our chain.

## Appendix E: Entanglement entropy and central charge fits

### 1. Entanglement entropy

In this section we present more results on the entanglement entropy calculations in the middle of the chain $S(L/2)$ and provide a few more details.

In the main text we focused on entanglement entropy as a function of filling $n$, $\lambda$, and the $\mathbb{Z}_2$ electric field $h$. Here we also present the results for the $\mathbb{Z}_2$ LGT when the same data is plotted as a function of the chemical potential $\mu$, see the middle plots in Fig. 14. For the zero electric field $h = 0$, a clear qualitative change in the entanglement entropy can be observed at $\mu = \pm 2t$. Such change can be seen for both $\lambda > 0$ and $\lambda < 0$ albeit with the difference in the absolute magnitude of the entanglement entropy, see also discussion in the main text. The fact that we see a qualitative change at $\mu = \pm 2t$ shows that the simulated model (D1) for $h = 0$ indeed resembles the behavior of the generic one-dimensional superconducting model (A7), where a topological trivial to non trivial transition occurs for $\lambda \neq 0$ at $\mu = \pm 2t$ [28, 35].

Similar results can also be seen for the non-zero values of the $\mathbb{Z}_2$ electric field with two differences: the transition

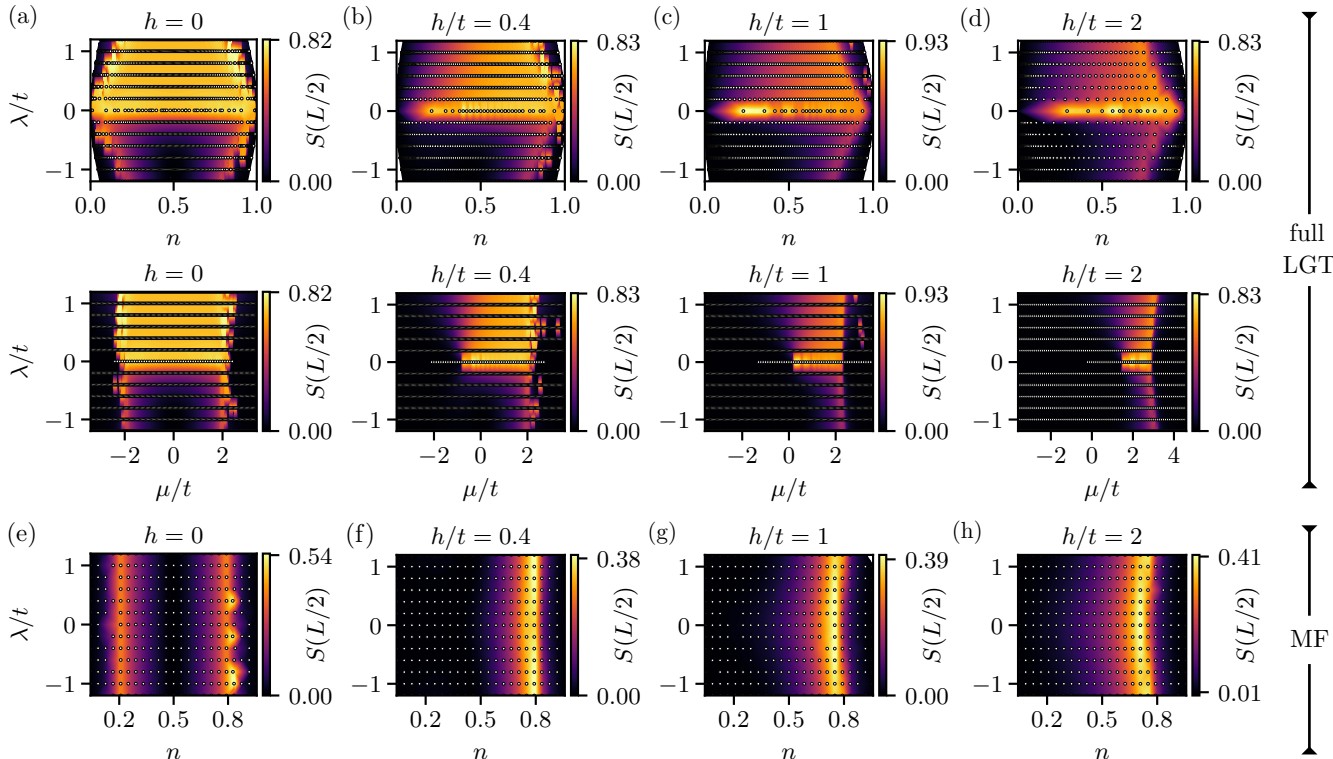

FIG. 14. Entanglement entropy, $S(L/2)$ as a function of $\lambda$ for different $\mathbb{Z}_2$ electric field values. $\mathbb{Z}_2$ LGT results are presented in sub-figures (a)–(d) where the top row shows the entanglement entropy as a function of filling $n$ and $\lambda$, for $h = 0$, $h/t = 0.4$, $h/t = 1$, and $h/t = 2$, respectively. The second row of (a)–(d) shows the same results but as a function the chemical potential, instead of filling $n$. The mean-field theory entanglement entropy (in the gauge sector) as a function of filling $n$ and $\lambda$ for $h = 0$, $h/t = 0.4$, $h/t = 1$, and $h/t = 2$ are shown in the third row labeled (e)–(h), respectively.

at $\mu = -2t$ disappears or at least the change in behavior of the entanglement entropy is smooth, and the transition at $\mu = +2t$ is slightly shifted to a higher value, as a function of $h$, see Fig. 14(b)–(d).

In Fig. 14(e)-(g) we also plot the entanglement entropy value in the middle of the chain for the mean-field theory Eq. (9), for more values of the $h$ as in the main text. We observe that the sole band of high entanglement entropy at $n \approx 0.8$ only slightly shifts to a lower value of $n$ with increasing $h$, see Fig. (14)(h).

In all results above we see some outlying data points when $\mu$ is either very low or very high. We assume that these are some poorly converged data points in trivial states where DMRG got stuck and took a bigger bond dimension than actually needed. In addition we exclude the results where some points in the profile $S(x)$ were missing and when the difference between the target filling and the obtained filling $n(p)$ in the last step was larger than one percent.

In addition, the phase boundaries are not completely smooth as a function of $\lambda$. We again attribute this to the fact that DMRG sometimes struggles at the quantum phase transition, and the convergence is slightly worse. Despite this complication, we obtain a correct qualitative value of $\mu = \pm 2t$ which means that our calculations are accurate.

### 2. Fits of the entanglement entropy

Here we provide more details on the fits of the entanglement entropy $S(x)$. As already stated in the main text we use a trick where we normalize the entanglement entropy as [44, 45]

$$\tilde{S}(x) = \frac{S(x)}{n(x)} n, \tag{E1}$$

in order to reduce the effect of oscillations. This is in particular useful when considering the $\lambda = 0$, where the charge number is conserved and we observe strong Friedel oscillations.

For $h = \lambda = 0$ we in fact take into account the particle-hole symmetry and modify the expression in Eq. (5) for $n > 0.5$ to

$$\tilde{S}(x) = \frac{S(x)}{1 - n(x)} (1 - n), \tag{E2}$$

which increases the quality of the fits even more.

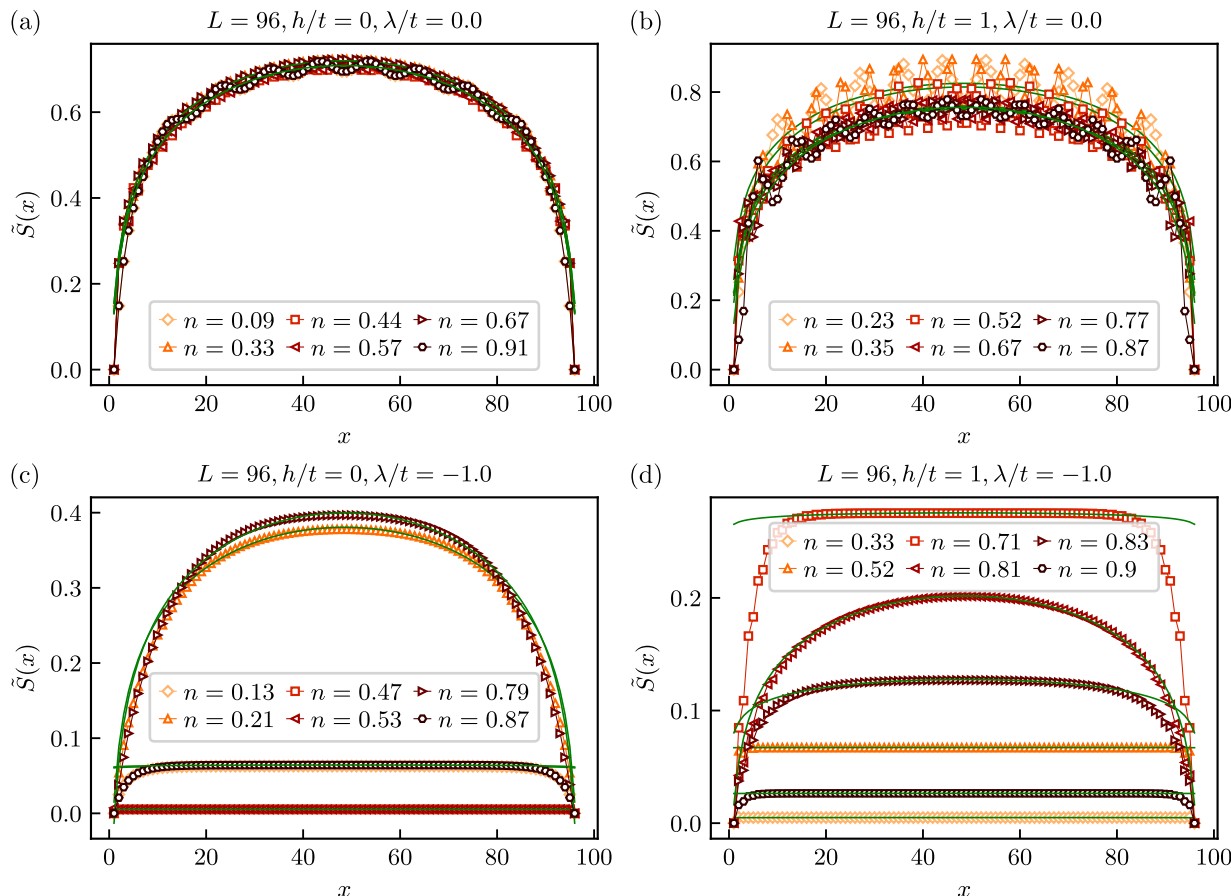

FIG. 15. Entanglement entropy $\tilde{S}(x)$ normalized by the local density $n(x)$ in the $\mathbb{Z}_2$ LGT Eq. (1). (a) The central charge does not change with filling $n$ for free particles at $h = \lambda = 0$ and is indeed close to $c = 1$, which signals a gapless Luttinger liquid of free partons. (b) The central charge also does not change with filling $n$ in the confined phase $h/t = 1$ when $\lambda = 0$. The value of the central charge is again close to $c = 1$, which signals a Luttinger liquid of mesons. Note that the oscillations in $\tilde{S}(x)$ are more pronounced than in the $h = 0$, where the trick with the local density normalization almost completely eliminates oscillations. (c) The shape of the entanglement entropy $\tilde{S}(x)$ for $h = 0$ and $\lambda/t = -1$ changes from a flat profile to a curved profile at the transition from trivial to topological state. The central charge value is lower, and is estimated to be $c = 0.5$. Our fit result yielded $c = 0.64 \pm 0.01$ for $n = 0.21$ an $c = 0.72 \pm 0.01$, with errors estimated from the variance of the fit parameter. (d) Similar change is observed also for $h/t = 1$ and $\lambda/t = -1$, with the only difference that the flat plateau of the entanglement entropy rises with filling until $n \approx 0.75$, where it acquires some curvature around $n \approx 0.8$, and becomes flat again and decreases for even higher fillings.

We use the CFT formula Eq. (4) from the main text [41–43]

$$S_{\text{CFT}}(x) = S_0 + \frac{c}{6} \log \left[ \left( \frac{2L'}{\pi} \right) \sin \left( \frac{\pi x}{L'} \right) \right]. \quad \text{(E3)}$$

to fit our normalized entanglement entropy $\tilde{S}(x)$ and extract the central charge $c$. The second fit parameter $S_0$ is non universal and we do not analyse it further. Finally, we keep the length constant at the actual system length $L'$.

We present some of the typical results of the normalized entanglement entropy profiles $\tilde{S}$ with the fits in Fig. 15. We note that the data points across different fillings, for $\lambda = 0$ in the deconfined $h = 0$ and confined regime $h/t = 1$ exhibit similar curvature, which is in line with the fact that the system is critical and the central

charge is always $c = 1$, see Fig. 15(a) and (b). We note that the Friedel oscillations are not mitigated as well as for the $h = 0$ regime.

For $\lambda \neq 0$ regime we see that $\tilde{S}(x)$ profiles are indeed almost completely flat for $15 < x < L - 15$ for the regime away from criticality. When the system approaches the transition we note that the entanglement entropy obtains a signature curvature. As explained in the main text the central charge is lower at $c = 0.5$ [46]. We note that our fit results are slightly worse as we frequently overshoot the correct value, see Fig. 15(c) and (d). However the qualitative picture is correct.

We note that fit result for the mean-field theory Eq. (11) look similar to the results for the exact model when $\lambda \neq 0$ and as a result we do not present the details of those fits here.

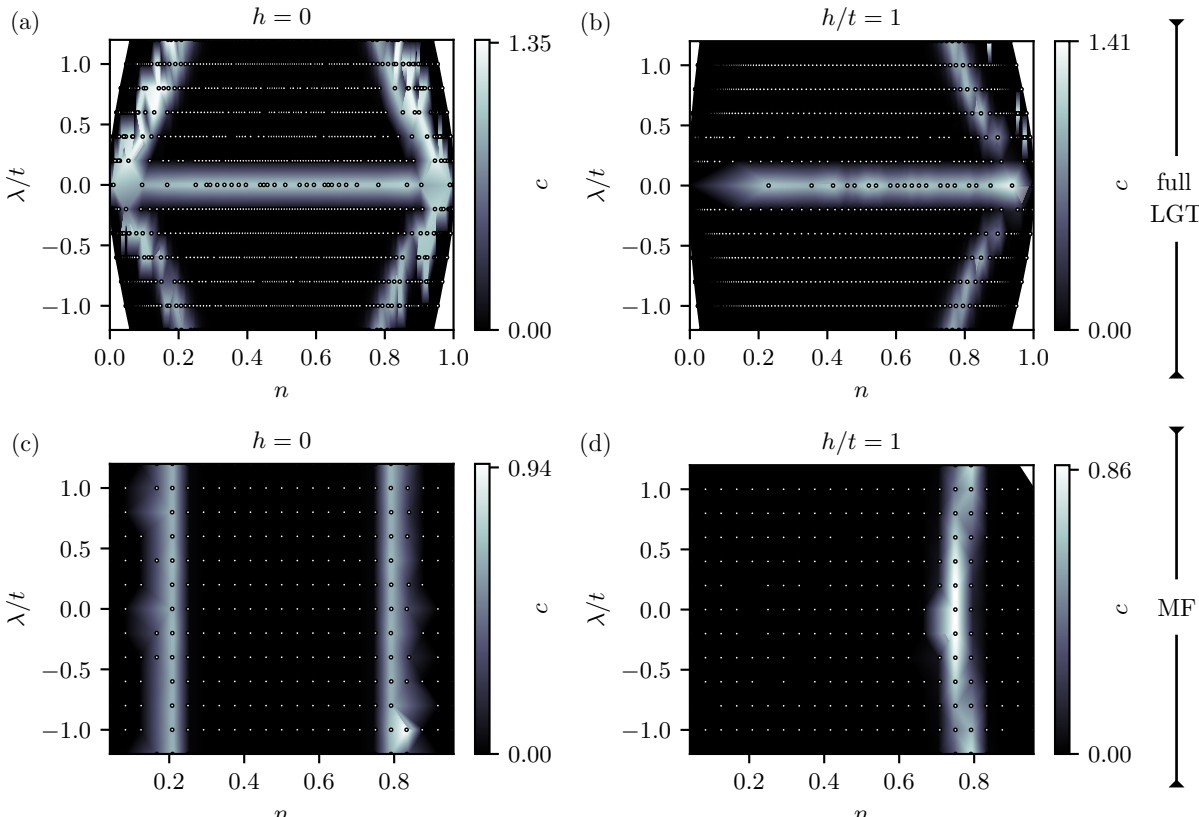

FIG. 16. Central charge results extracted by fitting the entanglement entropy $\tilde{S}(x)$ with Eq. (E1) and Eq. (E2) for the $\mathbb{Z}_2$ LGT Eq. (1) and the mean-field theory Eq. (9). (a) $\mathbb{Z}_2$ LGT results for zero $\mathbb{Z}_2$ electric field $h = 0$. Without discarding fit results with lower quality, we obtain a more pronounced diagonal lines as a function of $n$ for $\lambda \neq 0$, when the $\mathbb{Z}_2$ field is zero $h = 0$. This agrees with the argument that the entanglement entropy results become more sensitive when the system approaches the criticality. (b) When the $\mathbb{Z}_2$ field is non-zero $h/t = 1$, the $\mathbb{Z}_2$ LGT results show only one line as a function of $n$ for $\lambda \neq 0$. Without discarding lower quality fits, we also see some outliers for $n \geq 0.8$. (c) Mean-field theory for $h = 0$, exhibits two vertical lines related to the transition between the topological and trivial states. Note that also here we present data points coming from fits with lower quality, which results in higher maximum value for $c$. (d) The results for the mean-field theory $h/t = 1$, show only one transition at $n \approx 0.8$, we note that the maximum value of $c$ is similar to the main text results.

We furthermore note that we only fitted the entanglement entropy profiles $\tilde{S}(x)$, in the range $15 < x < L - 15$ in order to correctly capture the flat profiles in the parameter regimes away from criticality. At this point we would once again like to stress the fact that the CFT formula is only applicable exactly at the quantum criticality meaning that we expect poor fit results for generic parameters away from criticality [41].

We also note that in the Figures in the main text we excluded fit results with too big errors, which we parameterize by the value of the covariant matrix element, i.e, the variance related to the central charge $c$. We generally excluded results with errors bigger than $\Delta c = \pm\sqrt{0.01}$, for $h = 0, \lambda \neq 0$, and $\Delta c = \pm\sqrt{0.05}$, for $h/t = 1$. For the mean-field calculations we discarded errors bigger than $\Delta c = \pm\sqrt{0.02}$. Here we defined the error as the standard deviation, which is simply the square root of the variance,

obtained from the diagonal element of the covariance matrix. In every figure we also excluded the results, where some data points of $S(x)$ were missing. For the mean-field theory calculations we also discarded calculations where the difference between the obtained filling $n(p)$ after the last step described in Appendix D differed from the target filling $n$ by more than one percent.

The results without the above mentioned post selection criteria, and without the normalization of the entanglement entropy with the local density are presented in Fig. 16. We still excluded results where some data points of $S(x)$ were missing. Here we see that the central charge of very high values of $c$ occurred also in the regions where the state is trivial and furthermore quite far away from the actual quantum transition. We attribute these point to the same outliers encountered in the entanglement entropy central value $S(L/2)$, where the calculations simply

did not completely converge. The shape of the peaks in such $S(x)$ also appear somewhat artificial. They somehow resemble shapes where the system contains a single charge or a single hole. This means that DMRG might jump into a sector with a single charge or hole instead of the vacuum or a fully filled chain which we should obtain.

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
