# Peer review of "Mean-field theory of 1+1D $\mathbb{Z}_2$ lattice gauge theory with matter"

_SciPost Physics_

## Round 1 · Referee Report · Anonymous (Referee 1) · 2024-6-19

Report

The authors study a model of spinless hard-core bosons on a 1D lattice with a p-wave pairing term and coupled to a dynamical Z2 gauge field. By Jordan-Wigner transformation, this model is equivalent to the same model with hard-core bosons replaced by fermions, that was studied previously in Ref. 28. Using mean-field theory and DMRG simulations, the authors vary model parameters such as filling, strength of pairing term, electric string tension and map out the ground-state phase diagram, largely confirming the results obtained in Ref. 28. In addition to the entanglement entropy, the authors also compute string-length histograms/net electric field polarization and boson Green’s function to diagnose (de)confinement.

The paper is clear, detailed, and well written. The work presents interesting results that enrich our understanding of (discrete) lattice gauge theories (LGTs) with matter. This is a topic of enduring interest that has picked up speed in recent years, largely motivated by new experimental prospects for the realization of LGTs in synthetic matter (e.g. cold atoms and quantum computers).

I would like to offer the following comments/questions, in order of decreasing importance:

  1. My main concern regarding possible publication in SciPost Physics is the novelty/importance of the work. First, a large fraction of the manuscript is devoted to confirming results previously obtained in Ref. 28 (if not quantitatively, at least qualitatively). I believe most (if not all) the phases appearing in the phase diagrams presented here had already been found, and in the same model, by Ref. 28. Second, mean-field methods have a long history in LGT (see, e.g., Drell, Quinn, Svetitsky, Weinstein, PRD 19, 619 (1979); Boyanovsky, Deza, Masperi, PRD 22, 3034 (1980); Horn, Weinstein, PRD 25, 3331 (1982); and probably several others), but the authors present their mean-field approach for Z2 gauge theory as an entirely new idea. Regarding both points, which contributions would the authors say constitute important, conceptually new results? The authors should state this very clearly and also explain the relation to previous work more carefully (with appropriate citations as necessary).

  2. In the mean-field Hamiltonian (8), the authors drop the electric field term $<\tau^x_{i,j}>$ on account that it is a constant energy offset. I am concerned about this because this constant can take different values in different variational states, and thus potentially affect the phase diagram (which should reflect which state has the absolute lowest energy).

  3. The authors argue that the antiferroelectric state $<\tau^x_{j,j+1}> \sim (-1)^j$ is stable upon turning on a nonzero h term, while the ferroelectric state $<\tau^x_{j,j+1}> \sim$ const. is not. This confuses me for the following reason. For h=0, the model (1) actually has a global Z2 symmetry under reversal of all the $\tau^x$ electric fields, which is generated by a global Wilson loop $W=\prod_j \tau_{j,j+1}^z$ around the entire lattice. (Note that this symmetry also preserves the Gauss’ law constraint.) First, the authors should explicitly mention this global symmetry of the model after they introduce it, especially since they often constrast the h=0 and $h\neq 0$ cases throughout the paper. Second, if $h\neq 0$, that global Z2 symmetry is explicitly broken. Thus, it would seem to me that neither the ferroelectric nor the antiferroelectric state correspond to genuine spontaneously broken symmetries when h\neq 0. How should one then understand the stability of the antiferroelectric phase? Is it because it also breaks translation symmetry while the ferroelectric state (and the SPT phase) does not? (By the way, the authors may consider using “ferroelectric” and “antiferroelectric” instead of FM and AFM when discussing those phases.)

  4. I am confused why the gauge-invariant Green’s function (6) does not decay exponentially at h=0 when the pairing term \lambda is nonzero. As the authors show in Appendix A, in that limit the model simply maps onto the “ungauged” Kitaev chain (A4), and the 2-point function (6) maps onto the ordinary $$ 2-point function of the gauge-invariant $b,b^\dagger$ bosons. The Kitaev chain describes a gapped superconductor, thus I would expect the Green’s function to decay exponentially even in the topological phase.

  5. In Fig. 4(c,d) and Fig. 9(c,d), although this is just a suggestion, I wonder why the authors did not use a log-linear plot to more clearly demonstrate the exponential decay.

  6. Fig. 11 and the Sec. IV.D that discusses it are somewhat confusing because they give the impression that the mean-field theory becomes exact in the h=0 limit, which is clearly not true since for example, the phase boundaries obtained from the mean-field entanglement entropy (Figs. 7-8) do not match those from the full theory (Figs. 2-3). The authors should consider whether Fig. 11 is really necessary, since it only represents “partial” phase diagrams (in gauge/matter sectors separately).

  7. At the end of Sec. V.B, the authors write that “the mean-field theory correctly captures the qualitative features of the full Z2 LGT, especially when $\lambda\neq 0$, when it starts to match quantitatively.” The latter statement contradicts Figs. 13(a) and (c), which show that the $\lambda$ dependence of P is not captured by the mean-field theory.

  8. The most obvious “failure” of the mean-field theory is its inability to capture the filling dependence of the confinement phase boundaries. Can the authors speculate why mean-field theory does not capture this?

  9. At the beginning of Sec. V.A: “… for different values of the Z2 electric field in Fig. 12.” I assume they mean different values of the pairing term $\lambda$? Relatedly, in the caption of Fig. 12, the authors could perhaps point out that the legend in (b) applies to both panels.

  10. In several places (especially Appendix E, but also elsewhere), the authors use the expression “the Z2 electric field h”. h itself is not the electric field, so they should write “the Z2 electric field term h” or “the Z2 electric string tension h”.

  11. In Appendix A, the subscripts j,j+1 are not enclosed in <…> while they are elsewhere in the text.

  12. Caption of Fig. 1: “anti-stings” --> “anti-strings”

Recommendation

Ask for major revision

  • validity: -
  • significance: -
  • originality: -
  • clarity: -
  • formatting: -
  • grammar: -

Author:  Matjaž Kebrič  on 2025-11-06  [id 6001]

(in reply to Report 1 on 2024-06-19)

We thank the referee for their report.
Please find our reply to Report 1 in the attached file.

Attachment:

Ref1Reply.pdf

---

## Round 1 · Referee Report · Anonymous (Referee 2) · 2024-7-1

Report

Summary:

The paper studies the $(1+1)D$ quantum version of the Abelian Higgs model with discrete, Abelian gauge group, modeled by hardcore-bosons coupled to $Z_2$ gauge fields on the lattice, including a superconducting pairing term.
The authors present simulation results of the model, obtained via DMRG, and map out the phase diagram in the space of electric field strength $h$, filling $n$ and coupling constant $\lambda$, thus re-deriving to a large extent the results on the phase diagram of Borla et al, 2021 (Ref.28 in the current version).
The work subsequently sheds more light on the confinement aspect of the theory using as observables the gauge invariant two-point function and string-length histograms.
As actual novelty the authors solve the model using a mean-field, self-consistent approach, starting with a product-state ansatz for the wavefunction. The phase diagram obtained in mean-field theory is subsequently compared to the full theory finding good agreement in many but not all aspects.

General remark:

The paper is clearly written and gives a good account of the theory underlying the model. Ultimately, the work lives up to its title and delivers the mean-field solution to the theory under study. In addition, the phase diagram of the full theory is revisited with different observables, thus shedding some more light on the microscopic mechanisms at work. The authors discuss at length these aspects of the phase diagram found in the full theory via DMRG and using a mean-field approach. Even though the mean-field, variational approach for LGTs in the Hamiltonian formalism is a well-known method, the paper represents original research and has a sufficient degree of novelty.

Requested changes

1) The dynamical confinement mechanism for the theories with $(h=0, \lambda\neq 0)$ and $(h>0)$ seems to be different. In particular, the authors offer an explanation for the theory at low filling and $(h=0, \lambda\neq 0)$ where partons are basically created in pairs, separated by one unit of the lattice spacing which thus results in the corresponding string-length histogram which is peaked at $l=1$. What about at high filling for the same theory $(h=0, \lambda\neq 0)$?

2) Sect. 3B.1: The entanglement entropy $S(x)$ is used here but never formally defined in the work.

3) Sect. 3B.1, in the discussion relating the theories at $\lambda=1$ and $\lambda=-1$: "Indeed, by a gauge transformation ..." This should read "unitary transformation", as discussed in the appendix, since the model is only gauged under $Z_2 = {1, e^{\pm i\pi}}$.

4) Sect. D, towards the end of the section: "... in addition to the deconfined meson (parton)..." What is meant is the confining phase or meson LL.

5) Fig.4 and Fig.9: Displaying the correlators with log-linear scale would perhaps be beneficial.

Recommendation

Ask for minor revision

  • validity: -
  • significance: -
  • originality: ok
  • clarity: good
  • formatting: -
  • grammar: -

Author:  Matjaž Kebrič  on 2025-11-06  [id 6002]

(in reply to Report 2 on 2024-07-01)

We thank the referee for their report.
Please find our reply to Report 2 in the attached file.

Attachment:

Ref2Reply.pdf

---

## Round 2 · Referee Report · Anonymous (Referee 1) · 2025-11-27

Report

The authors have addressed my comments/questions satisfactorily. In my opinion the paper is now suitable for publication in SciPost Phys.

Recommendation

Publish (meets expectations and criteria for this Journal)

---

## Round 2 · Referee Report · Anonymous (Referee 2) · 2025-12-21

Report

The requested changes were implemented and clarifications were added. I recommend the article for publication.

Recommendation

Publish (meets expectations and criteria for this Journal)

---

## Round 2 · Author Response

We would like to thank the editor and the referees for their time and a thorough analysis of our manuscript. We are grateful for the many helpful comments. We incorporated all of the suggestions and comments made by the referees. A detailed response to the referee reports can be found in the attached files of the previous submission.

---

## Round 2 · List of Changes

The list of changes can be seen in the referee replies.

---

## Editorial Decision

voting_in_preparation